# Obesity-associated microbiomes instigate visceral adipose tissue inflammation by recruitment of distinct neutrophils

Dharti Shantaram [1,8], Rebecca Hoyd[2,8], Alecia M. Blaszczak[1,8], Linda Antwi[2], Anahita Jalilvand[1], Valerie P. Wright[1], Joey Liu[1], Alan J. Smith[1], David Bradley[1], William Lafuse[3], YunZhou Liu [2], Nyelia F. Williams [2], Owen Snyder[2], Caroline Wheeler[2], Bradley Needleman[4], Stacy Brethauer[4], Sabrena Noria[4], David Renton[4], Kyle A. Perry[4], Prabha Nagareddy[5], Daniel Wozniak [3], Sahil Mahajan[3], Pranav S. J. B. Rana [3], Maciej Pietrzak[6], Larry S. Schlesinger [7], Daniel J. Spakowicz [2] ✉ & Willa A. Hsueh [1] ✉

Neutrophils are increasingly implicated in chronic inflammation and metabolic disorders. Here, we show that visceral adipose tissue (VAT) from individuals with obesity contains more neutrophils than in those without obesity and is associated with a distinct bacterial community. Exploring the mechanism, we gavaged microbiome-depleted mice with stool from patients with and without obesity during high-fat or normal diet administration. Only mice receiving high-fat diet and stool from subjects with obesity show enrichment of VAT neutrophils, suggesting donor microbiome and recipient diet determine VAT neutrophilia. A rise in pro-inflammatory CD4+ Th1 cells and a drop in immunoregulatory T cells in VAT only follows if there is a transient spike in neutrophils. Human VAT neutrophils exhibit a distinct gene expression pattern that is found in different human tissues, including tumors. VAT neutrophils and bacteria may be a novel therapeutic target for treating inflammatory-driven complications of obesity, including insulin resistance and colon cancer.

Obesity is an epidemic within the United States, as over 70% of the adult population have an overweight (BMI 25 to <30 kg/m²)[1] or obese (BMI ≥ 30 kg/m²)[1] condition. Obesity increases the risk of numerous complications, including insulin resistance, type 2 diabetes (T2D), non-alcoholic steatohepatitis, Alzheimer's disease, cardiovascular disease, stroke, sleep apnea, poor wound healing, cancer, and other comorbidities[2]. Inflammatory processes within adipose tissue (AT) play a fundamental role in developing chronic inflammation, leading to these obesity-related comorbidities[3,4]. In lean mice, adipose tissue (AT) inflammation is repressed by an orchestrated interaction between immunosuppressive CD4+ regulatory T cells (Tregs)[5], anti-inflammatory M2-like macrophages[6–9], type 2 innate lymphoid (ILC2) cells[10,11], and eosinophils[12]. In contrast, obesity is associated with a dramatic increase in AT proinflammatory CD4+ type 1T helper (Th1)

[1]Diabetes and Metabolism Research Center, Division of Endocrinology, Diabetes & Metabolism, Department of Internal Medicine, Wexner Medical Center at The Ohio State University, Columbus, OH 43210, USA. [2]Pelotonia Institute for Immuno-Oncology at The Ohio State University Comprehensive Cancer Center–Arthur G. James Cancer Hospital and Richard J. Solove Research Institute, Columbus, OH 43210, USA. [3]Department of Microbial Infection and Immunity, The Ohio State University, Columbus, OH 43210, USA. [4]Center for Minimally Invasive Surgery, Department of General Surgery, The Ohio State University, Columbus, OH 43210, USA. [5]Department of Internal Medicine, Cardiovascular Section University of Oklahoma Health Sciences Center (OUHSC), Oklahoma City, OK 73117, USA. [6]Department of Biomedical Informatics, The Ohio State University, Columbus, OH 43210, USA. [7]Host Pathogen Interactions Program, Texas Biomedical Research Institute, San Antonio, TX 78227, USA. [8]These authors contributed equally: Dharti Shantaram, Rebecca Hoyd, Alecia M. Blaszczak. ✉e-mail: Daniel.Spakowicz@osumc.edu; Willa.Hsueh@osumc.edu

cells[13], cytotoxic CD8+ T cells[14], and M1-like macrophages[6], and a drop in Tregs[15–17], resulting in chronic low-grade inflammation[18–24]. However, an often overlooked immune cell within the AT microenvironment is the neutrophil, a key mediator of the innate immune response[25].

In mice, AT neutrophils appear to instigate AT inflammation and insulin resistance in diet-induced obesity[26–28]. Talukdar et al. showed an increase in neutrophil AT infiltration after three days on a high-fat diet (HFD) with a return to baseline at day 28, suggesting a neutrophil role early in AT adaption to HFD. Knockout of neutrophil elastase, over-expression of the neutrophil elastase inhibitor α1-antitrypsin, or the administration of an elastase inhibitor attenuated AT inflammation and the development of insulin resistance[24,26]. Loss of another key granular heme protein, myeloperoxidase, resulted in attenuation of weight gain, VAT immune cell infiltration, and insulin resistance[27]. Thus, in mice VAT neutrophils can instigate HFD-induced AT inflammation to disrupt systemic metabolism.

In this investigation, we observed the enrichment of VAT neutrophils in human obesity, identified a transcriptional profile distinct from PB neutrophils, and found that their presence may be related to bacterial translocation from the gut, depending on diet and type of bacteria. We then identified a VAT-isolated neutrophil (VIN) signature and found this signature to be widespread and associated with overall survival in obesity-related cancer, colon adenocarcinoma.

## Results

### VAT neutrophils are increased in individuals with obesity compared to lean individuals

Ninety-six consecutive patients were recruited for this prospective investigation. Subjects with an obese ($n = 82$) vs. lean ($n = 14$) condition had significantly higher circulating insulin, leptin, triglycerides, decreased adiponectin, and were older (Table 1). Homeostatic model assessment for insulin resistance (HOMA-IR), an indicator of insulin action, was higher in individuals with obesity. Subjects with obesity also had higher levels of plasma zonulin, a controversial marker of intestinal permeability[29,30], which was supported by increased

circulating levels of lipopolysaccharide (LPS) binding protein (LBP) that indicates bacterial presence. However, plasma LPS levels were not different.

VAT of subjects with obesity contained a higher abundance of neutrophils, measured as a percent of CD45+ cells in stromal vascular fraction (SVF) assessed by flow cytometry analysis (Fig. 1A). Neutrophil abundance was positively associated with insulin resistance (HOMA-IR) (Fig. 1B). Gating strategy is shown in Supplementary Fig. 1. Adipocyte gene expression revealed increased *leptin (LEP)*, decreased *adiponectin (ADIPOQ)*, and an increase in numerous proinflammatory mediators including interleukin-1beta (*IL1B*), NLR family pyrin domain containing 3 (*NLRP3*), tumor necrosis factor (*TNF*), and the neutrophil chemoattractant interleukin-8 (*IL8*) (Fig. 1C), confirming inflamed adipose tissue. The percentage of VAT neutrophils correlated with adipocyte *IL8, IL1B, NLRP3,* and *LEP* gene expression (Fig. 1D–H). Taken together, these data link VAT neutrophils with insulin resistance and suggest that increased expression of adipocyte *NLRP3/IL1B* and *IL8* may be involved in recruiting neutrophils to VAT. Comparatively, LPS added to cultured human adipocytes increased IL8 expression nearly 200-fold, with lesser increases in *CXCl2* and *IL1B*, and decreased adiponectin by 50% (Supplementary Fig. 2). Histology of VAT obtained from patients with obesity suggested neutrophils were primarily marginated within the vasculature, with occasional neutrophils within the adipose (Fig. 1I). In contrast, in lean patients, neutrophils were generally not identified in either location, suggesting that neutrophils are recruited to VAT primarily in patients with obesity. Gender differences impact neutrophils[31–33]; we found that female subjects with obesity VAT had more neutrophils than lean subjects. There was a suggestive difference in males that was not statistically significant, possibly because there were fewer male subjects (Supplementary Fig. 3).

### VAT samples contain bacteria

Because of the finding of increased circulating zonulin and LBP in patients with obesity compared to lean patients (Table 1) and the correlation of % VAT neutrophils with plasma LPS in 69 samples (Fig. 2A), we next looked for the presence of bacteria in surgical VAT samples collected under sterile conditions prior to any gut manipulation. Amplification of the 16S bacterial ribosome under sterile conditions identified bacteria from over 13 phyla in VAT from lean patients ($n = 7$) and patients with obesity ($n = 10$), while sequenced negative control samples ($n = 2$) resulted in 9 amplicons that passed quality filters and were removed from all samples. The communities were heterogeneous. The dominant phylum was Firmicutes in most samples, but there was high variability between subjects (ranging from nearly 100% of the sequences to less than 10%) (Fig. 2B). A Bayesian estimator of community sources, comparing the observed bacterial community to reference communities from a variety of body and other sites, showed the most likely source of the community was the gastrointestinal tract, not the skin, mouth, or the environment (*e.g.*, soil) (Fig. 2C). In analyses of individual taxa, we identified the family *Streptococcaceae* and an unnamed genus in the family *Ruminococcaceae* as enriched in VAT from patients with obesity, while the order Bacilliales and genus *Marvinobryantia* were enriched in lean VAT (Fig. 2D).

### Gavage of HFD-fed mice with human fecal microbiome from subjects with obesity induces neutrophil accumulation in VAT

To further test the possibility of bacterial translocation driving neutrophil recruitment to VAT, we created human-microbiome avatar mice by depleting the mouse resident microbiome with broad-spectrum antibiotics and an antifungal (efficiency of depletion shown in Supplemental Fig. 4) and then re-colonizing with feces from human subjects with obesity ($n = 4$ biologically independent fecal donors) or without obesity ($n = 3$ biologically independent fecal donors) or saline control (Fig. 3A). Mice were then placed on either a high-fat diet (HFD) or normal chow for five days, at which time VAT and

## Table 1 | Patient demographics

|  | Lean ($n = 14$) | Obese ($n = 82$) |
|---|---|---|
| BMI (kg/m²) | 23.2 ± 0.4 | 47.5 ± 1.0*** |
| Age (years) | 54 ± 4 | 45 ± 1* |
| Gender | 5 F/9 M | 69 F/13 M |
| Presence of diabetes | 0/14 | 24/83 |
| Fasting glucose (mg/dL)ᵃ | 99 ± 5 | 86 ± 2* |
| Fasting insulin (µIU/mL)ᵃ | 6.8 ± 3.3 | 16.9 ± 2.6* |
| HOMA-IR scoreᵃ | 1.8 ± 0.9 | 3.1 ± 0.4 |
| Total cholesterol | 7.0 ± 1.6 | 9.3 ± 0.4 |
| Triglycerides | 79 ± 8 | 193 ± 16ᵗ |
| LDL | 99 ± 23 | 100 ± 5 |
| HDL | 54 ± 4 | 47 ± 2 |
| ALT | 21 ± 5 | 28 ± 11 |
| AST | 23 ± 3 | 25 ± 8 |
| Plasma adiponectin (ng/mL) | 13494 ± 2552 | 8338 ± 589** |
| Plasma leptin (ng/mL) | 20.8 ± 5.5 | 52.3 ± 3.3*** |
| Plasma zonulin (ng/mL) | 31.97 ± 14.14 | 71.59 ± 6.56* |
| Plasma LPS (pg/mL)* | 27.7 ± 4.8 | 27.4 ± 3.0 |
| Plasma LBP (ng/mL)* | 33.4 ± 5.5 | 44.0 ± 3.1** |

All values are expressed as mean ± SEM.
*BMI* body mass index, *HOMA-IR* homeostasis model assessment of insulin resistance, *LBP* LPS binding protein.
ᵗ$p < 0.1$, *$p < 0.05$, **$p < 0.01$, ***$p < 0.001$.
ᵃpatients with diabetes excluded from the analysis.

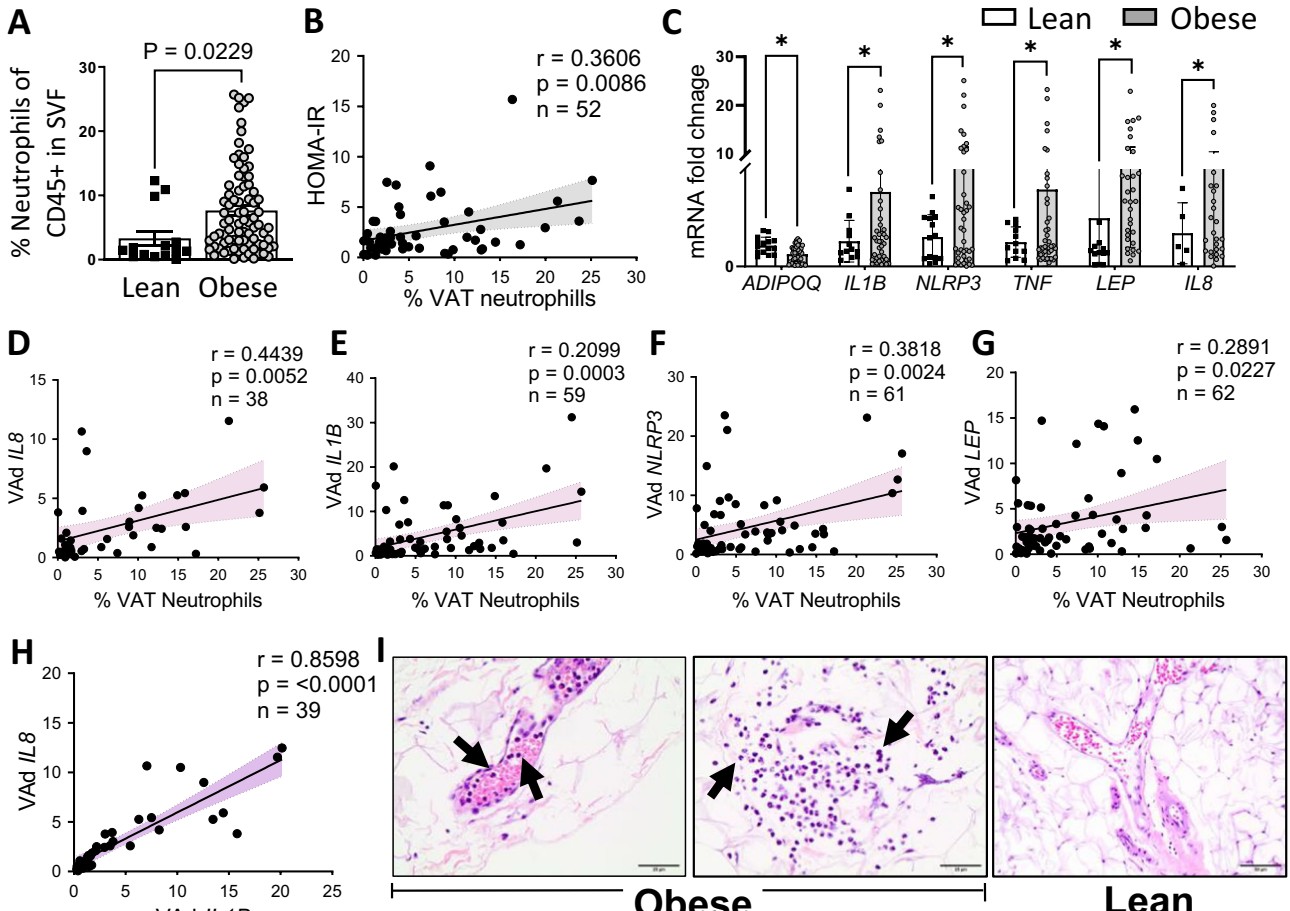

**Fig. 1 | Visceral adipose tissue (VAT) neutrophils are increased in subjects with obesity compared to lean subjects.** Neutrophil abundance is higher in **A** Visceral adipose tissue (VAT) from subjects with obesity ($n = 82$) compared to lean subjects ($n = 14$) AT assessed by flow cytometry of stromal vascular fraction (SVF) of VAT, which also **B** correlates ($n = 52$, Pearson correlation, with 95% CI) with homeostatic model assessment of insulin resistance (HOMA-IR). **C** mRNA expression in visceral adipocytes (VAd) assessed by qRT-PCR, more inflammatory genes expressed in VAd from patients with obesity ($n = 46$) compared to lean ($n = 16$), gene expression of **D** *IL8*, **E** *IL1B*, **F** *NLRP3* and **G** *LEP* (Leptin) all are positively correlated with the % VAT neutrophils (Pearson correlation, with 95% CI), with a strong correlation between **H** VAd *IL1B* and *IL8* gene expression. **I** Representative images of histological examination of neutrophils in VAT ($n = 6$ for obese and $n = 4$ for lean AT), within the vasculature, and upon infiltration into the obese adipose tissue (scale bar = 25 μm) in comparison to lean adipose tissue in which no neutrophils are present in the vasculature or the tissue (scale bar = 50 μm). Data represented as mean ± SEM, with a comparison between two groups using unpaired student's *t*-test and Pearson correlations with two-tailed analysis, *$p < 0.05$.

other tissues (brain, lung, liver, etc.) were harvested for immune cell analyses (gating of mouse neutrophils and macrophages shown in Supplementary Fig. 6). Ribosomal amplicon sequencing confirmed the presence of bacterial genomes in VAT, but not in other tissues (brain and liver), for all mice placed on HFD, regardless of the gavage (Fig. 3B). The saline gavage also showed greater bacterial load in VAT after HFD relative to a normal chow diet, suggesting that sufficient microbes remained in the gut to translocate to VAT. The most abundant microbes in VAT following the saline gavage included *Ralstonia pickettii, Delftia acidovorans, Rhodococcus erythropolis qingshengii sp5959*. This finding is consistent with bacteria levels in mouse stool collected throughout the experiment, where the total amount of bacteria was significantly reduced by antibiotics but not to zero (Supplementary Fig. 4A). Further, we confirmed gavage efficacy and engraftment, as measured by the similarity of mouse stool to the donor material at the end of the experiment (Supplementary Fig. 4B). Mice placed on a normal chow diet did not show increased bacteria in VAT, even in the context of a donor gavage from a subject with obesity (Fig. 3B). Flow cytometry analysis confirmed neutrophil enrichment in VAT, which was only observed with the combination of gavage from a

subject with obesity and HFD (Fig. 3C); neither the lean feces nor saline gavage resulted in enriched neutrophils, despite similar bacteria levels in VAT. Macrophages were not enriched in any condition (Fig. 3C). Thus, despite obese fecal gavage, neutrophils only increased when mice ingested HFD (Fig. 3D). Gene expression in SVF confirmed the flow cytometry results: expression of *Ly6g*, a neutrophil marker, was increased 10-fold in mice ingesting HFD and gavaged with feces from donors with obesity with no increase in macrophages using *Emr1* as a marker (Fig. 3E). In addition, the inflammatory effectors, *Il1b* and *Spp1*, were increased in mice on HFD gavaged with feces from donors with obesity but not in VAT from the other groups (Fig. 3E). Only one microbial taxon, the genus *Pseudomonas*, was enriched in the VAT from mice on HFD gavaged with fecal microbiome from individuals with obesity (Fig. 3F). Repeating the experiment with three other human donors with obesity, each of which led to increased neutrophil recruitment in VAT, revealed enriched members of the phylum Proteobacteria, including *Pseudomonas stutzeri* and both α- and β-*Proteobacteria* classes (Fig. 3G). There was no significant difference in the VAT microbe α-diversity or number of observed species (Supplementary Fig. 5). These results suggest that HFD leads to bacterial

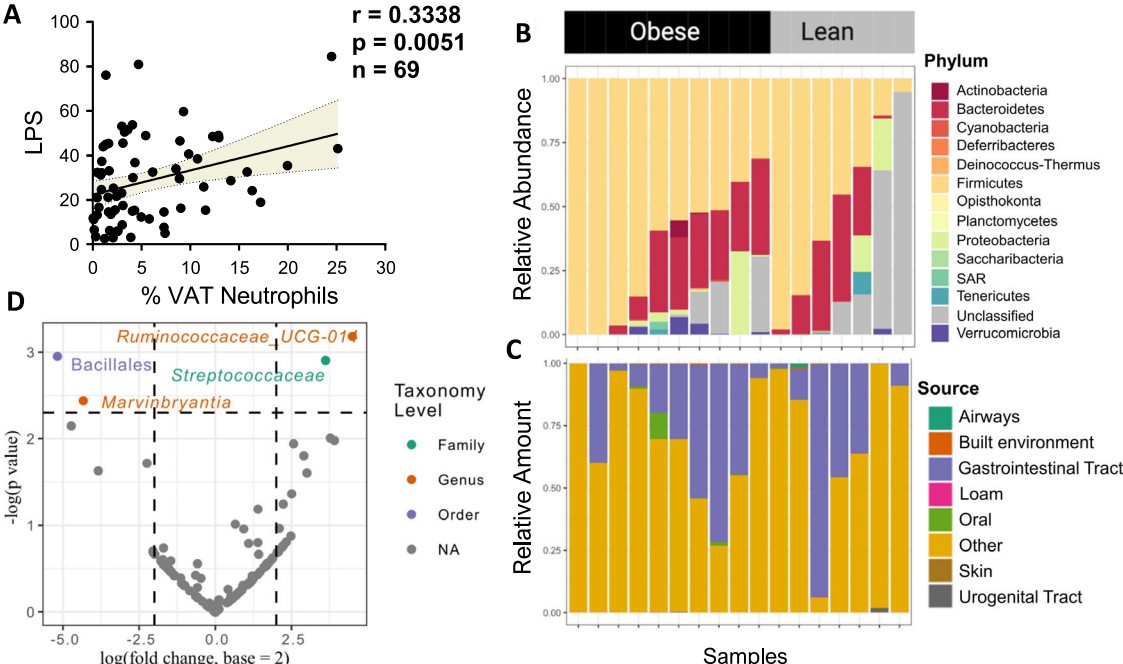

**Fig. 2 | Bacteria are present in adipose tissue and are different between VAT from subjects with obesity compared to lean subjects. A** VAT neutrophil abundance correlates with circulating LPS (n = 69, Pearson correlation, with 95% CI). 16S rRNA sequencing revealed that **B** the relative abundances of bacteria in individuals with obesity (n = 10) and lean (n = 7) individuals is similar at the level of the phylum, and **C** the most commonly identified source is estimated to be the gastrointestinal tract by Bayesian estimator of community source. **D** Differential expression analysis reveals several bacterial taxa showed differences between obese and lean VAT, with the family Streptococcaceae and a genus in the family Ruminococcaceae enriched in obese VAT, while the genus Marvinobryantia and the order Bacilliales were enriched in lean VAT. The data plotted are the −log(unadjusted *p*-value) and log(-fold-change, base = 2) as calculated with (DESeq2) using two-sided tests, using the same 17 samples as appear in (**B**) and (**C**).

infiltration into VAT and that the presence of certain bacteria drives neutrophil recruitment. Fecal donor demographics are summarized in Supplementary Table 1. Of note, there was a progressive increase in neutrophils as the donor BMI increased (Supplementary Fig. 8). Donor gender did not alter these results (Supplementary Fig. 9). Consistent with our observations in mice and previous data demonstrating that HFD decreases intestinal tight junctions[34–36], we administered two weeks of HFD to healthy, lean individuals as described[17], and 9 out of 11 subjects had a rise in plasma zonulin. Thus, HFD is an important driver of gut changes leading to VAT inflammation.

To determine the functional importance of neutrophils, we continued to assess AT inflammation in VAT following the neutrophil spike. After the five-day HFD, mice were placed on a chow diet and followed for 11 more days (Fig. 4A). Although the neutrophil spike subsided, there was a marked rise in CD4+ Tbet+ Th1 cells and a drop in CD4+ CD25+ Foxp3+ regulatory T cells (Tregs) in VAT, but no differences in macrophage numbers were detected (Fig. 4B–E, Supplementary Figs. 6 and 7 for gating strategy). To test whether neutrophils drove the T cell changes in these mice, we administered a neutrophil depletion antibody against Ly6G (αLy6G) at the start of the HFD for three days (Fig. 4F). The antibody efficiently eliminated neutrophils from blood, VAT, and spleen (Supplementary Fig. 10). Not only was the VAT neutrophil spike eliminated, but the rise in Th1 cells and drop in Tregs was prevented (Fig. 4G–J). There were no differences in body weights, and the mice given the antibody remained insulin sensitive (Fig. 4J–M). Taken together, the avatar mouse model demonstrates events involved in the instigation of HFD-induced VAT inflammation: HFD allows bacterial translocation from the gut with entry into VAT but not other tissues, which induces VAT neutrophil accumulation, subsequently leading to proinflammatory T cell changes that are commonly found in obesity in both humans and mice[15,17,37,38].

## VAT neutrophils are distinct from peripheral blood (PB) neutrophils

To determine whether VAT neutrophils were simply migrating peripheral blood (PB) neutrophils, the expression of isolated PB and VAT neutrophils was compared. PB neutrophils were isolated using a dextran ficoll gradient, and VAT neutrophils were isolated by flow sorting from the same subjects (n = 9 obese and n = 6 lean samples). Isolation of PB neutrophils by the dextran ficoll gradient or flow sorting did not change their gene expression (Supplementary Fig. 11). Transcriptome analyses revealed significant differentially expressed genes (DEGs) in VAT vs. PB neutrophils, with fewer differences in PB taken from patients with obesity vs. lean patients, and VAT neutrophils from patients with obesity vs. lean patients. Clustering by global expression showed distinct but overlapping clusters for each group, with separate clustering of the VAT and PB neutrophils from subjects with obesity (Fig. 5A). VAT neutrophils showed significantly different expression in members of the pathways for inflammation, chemotaxis, extracellular-matrix production, reactive oxygen species (ROS), growth factors, and apoptosis (Fig. 5B). All genes in the pathways shown are significantly upregulated. These data suggest that neutrophils persist within human VAT and are partially activated, as not all inflammatory genes were increased. Lastly, the increase in genes related to bactericidal activity and LPS induction is consistent with these neutrophils having come into contact with bacteria, LPS, or other bacterial products.

## Few transcriptomic differences between VAT neutrophils from lean subjects and subjects with obesity

A comparison of VAT neutrophils from lean subjects vs. subjects with obesity revealed that there were 749 differentially expressed genes between VAT neutrophils taken from subjects with and without obesity; 55 were enriched in VAT neutrophils from subjects with

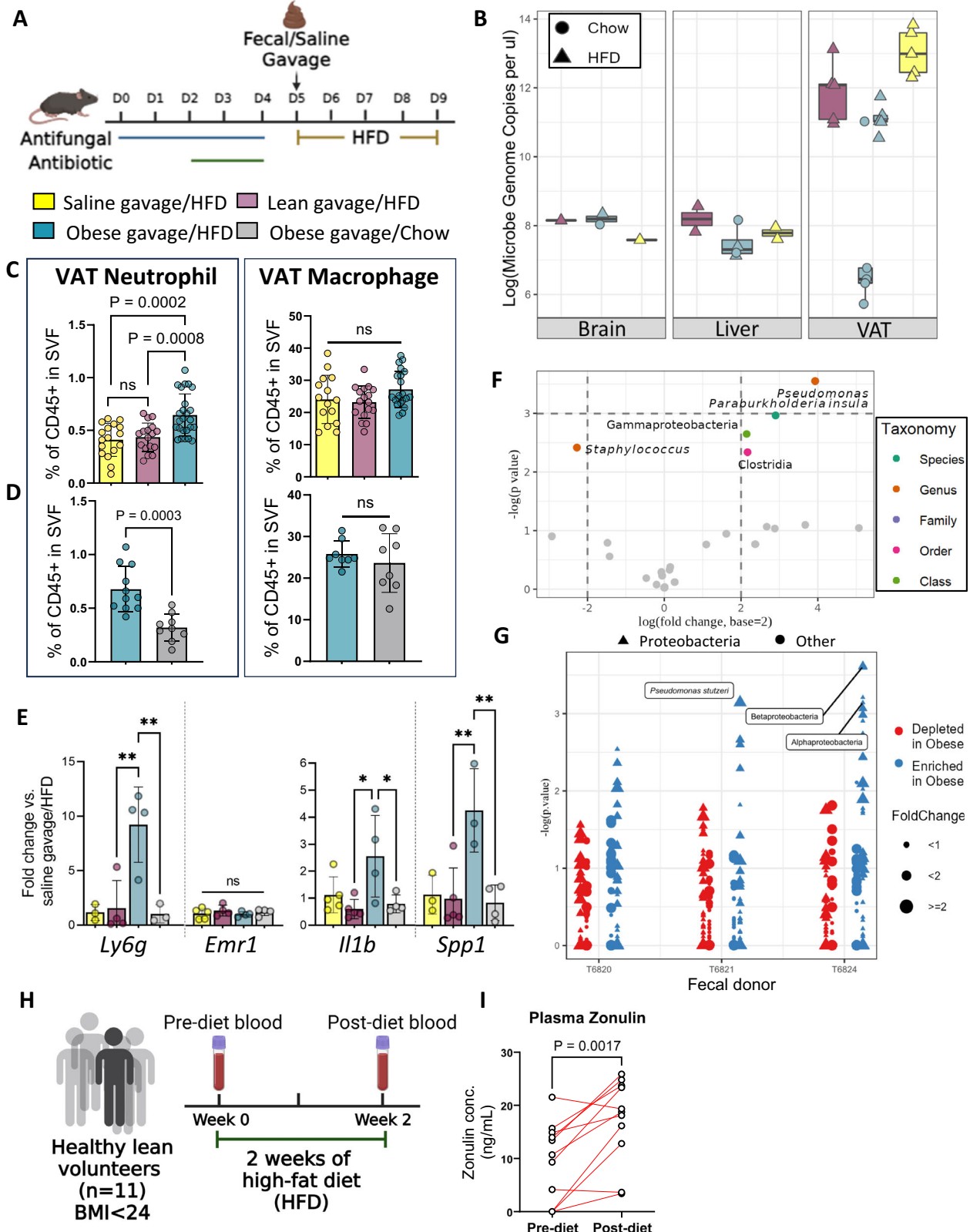

obesity, while 639 were enriched in neutrophils from lean subjects. Pathway analyses indicated 33 genes in the olfactory pathway and 20 genes in the PI3k pathway were enriched, while fewer genes were enriched in other KEGG pathways. No pathways were over-represented amongst the genes enriched in VAT neutrophils from patients with obesity. Given the relative similarity of the neutrophils from individuals with and without obesity (relative to VAT vs. PB

neutrophils), individuals with and without obesity were grouped in all further analyses.

## VAT-derived neutrophil activation is distinct from other activated neutrophil states

To explore whether the expression of neutrophils isolated from VAT, with the particular type of activation suggested by the pathway

**Fig. 3 | Recruitment of VAT neutrophils is dependent on mouse diet and fecal donor obesity. A** Stool from individuals with and without obesity and sterile saline was gavaged into mice (n = 5/ group). **B** Mice receiving obese stool showed increased microbial abundance in VAT when fed with HFD, but not normal chow (brain = 4, liver = 8, VAT = 20), **C** flow cytometry of VAT stromal vascular fraction (SVF) confirmed an increase in neutrophils but not macrophages in mice receiving obese gavage (n = 4 fecal donors), while no rise was observed with lean (n = 3 fecal donors) or saline gavage. **D** The rise in neutrophils after obese gavage (n = 2 fecal donors) is due to HFD, as this rise is not observed with chow (n = 6 mice/gavage). **E** VAT qRT-PCR confirmed an increase in neutrophil marker (*Ly6g*) and an unchanged macrophage marker (*Emr1*). Inflammatory genes *Il1b* and *Spp1* were upregulated in VAT after obese gavage and HFD, **F** Unique bacterial abundance in VAT of HFD-fed mice gavaged with obese compared to lean microbiome (obese = 1, lean = 1 stool, 5 mice/group). Displayed are log₂(fold-change) between obese and lean VAT and −log(unadjusted *p*-value) of microbial abundance association with obese vs. lean VAT in a univariate generalized linear model using a gamma

distribution. **G** Differences between VAT microbes in mice gavaged with lean or obese stool (n = 2 lean, 3 obese, 5 mice/group). Mice receiving obese stool were grouped by the donor and compared to mice receiving lean stool using univariate logistic regression. (unadjusted *p*-values). **H** In a clinical study[17], we collected plasma samples from 11 healthy volunteers pre- and post-HFD feeding for 2 weeks. **I** Enzyme-linked immunosorbent assay (ELISA) confirmed an increase in plasma zonulin levels post 2 weeks of HFD feeding. Data represented as mean ± SD compared by one-way ANOVA followed by post hoc Tukey's multiple comparisons for three groups, two groups compared using unpaired student's *t*-test with two-tailed analysis *$p < 0.05$, **$p < 0.01$, ***$p < 0.001$. ELISA was analyzed using a paired *t*-test with two-tailed analysis, **$p < 0.01$. Boxplot in (**B**) defined using first and third quartiles to bound the box. Minima is first quartile − 1.5 * inter-quartile range (IQR), and maxima is third quartile + 1.5 * IQR. Outliers shown as points. (**A** and **H** created with BioRender.com released under a Creative Commons Attribution-NonCommercial-NoDerivs 4.0 International license https://creativecommons.org/licenses/by-nc-nd/4.0/deed.en).

---

analysis (Fig. 5B), was similar to other reported observations of neutrophils, we retrieved publicly available datasets of neutrophils flow-sorted in a similar manner. We retrieved and uniformly processed neutrophil and control data from bronchial lavage and synovial fluid and PB neutrophils isolated from individuals who had recently exercised, were exposed to LPS, or were suffering from sepsis or active tuberculosis (Supplementary Table 2). In each case, we generated a set of DEGs, which were compared between these states and with VAT neutrophils. We used a 7-way modified Venn diagram-style analysis (UpSet), comparing the number of DEGs in each condition and how many were shared. Few significant DEGs were shared between conditions (the leftmost vertical bars were larger than those toward the right, where the interactions are shown in Fig. 5C). VAT neutrophils shared the most DEGs with the sepsis condition (red box, Fig. 5C). Notably, the expression of these genes was also most strongly correlated, further indicating the strongest similarity (Fig. 5C). The VAT neutrophil DEGs showed a significant positive correlation only with the endotoxin response to LPS, but a significant negative correlation with active tuberculosis, bronchial lavage, and exercise. However, ordination by global gene expression demonstrated that each activation type formed distinct clusters, including VAT (Fig. 5D). To put these differences in the context of typical neutrophil activation pathways, we generated signature scores for genes in the pathways described earlier (Fig. 5B). Again, VAT neutrophils showed similarities to other neutrophil conditions for some pathways, but in aggregate, it was distinct. For example, VAT neutrophils showed similar expression in the bactericidal activity pathway as sepsis neutrophils but significantly differed from sepsis neutrophils in inflammatory genes (Fig. 5E). In summary, we interpreted these data as indicating the distinct behavior of VAT neutrophils from any others we could compare to in the literature. The distinctiveness of VAT-isolated neutrophils implied the ability to detect expression signatures in the context of bulk tissues; therefore, we next sought to evaluate if the VAT-isolated neutrophil expression could be found elsewhere in the body.

### Creation of a custom-signature tool to identify VIN-type neutrophils within other transcriptome datasets

In order to estimate whether neutrophil characteristics observed in VAT are unique to that in tissue, we exploited the distinct expression profile of VAT neutrophils to create a custom-signature tool that could estimate the fraction of VAT-isolated neutrophil (VIN)-type neutrophils within bulk transcriptome datasets. This deconvolution method used support vector regression to select a set of genes that define each cell type and then fit the signature gene set to the bulk data to estimate abundance. We compared our VIN RNAseq with high-quality reference transcriptomes from 22 cell types, including macrophages, T cells, B-cells, dendritic cells, and others assembled into a matrix containing at

least three representatives of each cell type[39]. Support vector regression selected roughly 1000 genes in a signature matrix that most distinguished each cell subset and VINs (Fig. 6A). The VIN-type signature was distinguished, even from the most similar cell type (PB neutrophils), by proinflammatory mediators, including *IL8, IL1B GOS2, PLAUR, NAMPT, PTGS2, PPP1R15A, TREM1*, and *SOD2*. Several of these signature genes that strongly define VINs vs. their closest neighbor were validated by qRT-PCR using VAT and PB neutrophils collected from a validation cohort (n = 5, Fig. 6B, Supplementary Table 3). To test the performance of the VIN-type custom signature, we applied the tool to publicly available bulk-sequenced datasets from samples where we could anticipate few (blood) or many (whole AT) VIN-type cells to be present (accession numbers in Supplementary Table 4). RNAseq or microarray gene expression from over 70 blood and whole AT samples were deconvolved using the VIN-type custom signature. Uniformly, only VIN-type expression was identified within the adipose tissue samples, and no VIN-type expression was identified within the PB samples (Fig. 6C).

### Application of the custom-signature tool suggests VAT neutrophils are widespread throughout human tissues and correlate with cancer survival

Having validated the performance of the deconvolution signature tool, we next used it to explore the abundance of VIN-like neutrophils in diverse tissues. We leveraged large databases of healthy and diseased tissues using bulk-RNAseq data to estimate whether the distinct expression we observed in VINs is broader than in VAT alone. We deconvolved 17,382 RNAseq samples from 30 healthy tissues from the Genotype-Tissue Expression Project (GTEx) and found that the VIN-type signature was much more abundant than PB-type neutrophil signature in all tissues, except for blood and the (blood-rich) spleen (Fig. 6D).

Next, given the association between obesity and cancer, we applied the signature tool to publicly available tumor-biopsy RNAseq data to search for VIN-type neutrophils. We deconvolved 478 colon adenocarcinoma (COAD) tumor-biopsy RNAseq samples from The Cancer Genome Atlas (TCGA). As in the GTEx dataset, the VIN-type expression was much more abundant than PB-type expression in tumor tissues, with a median of 20% of deconvolved VIN-type vs <1% PB-type expression (Fig. 6E). Further, the relative abundance of VIN-type neutrophils, but not PB neutrophils, correlated with overall survival, whereas individuals with a higher relative abundance of the VIN-type signature exhibited longer overall survival (Fig. 6F). No such threshold could stratify survival with blood-like neutrophils. However, although the VIN-like expression was present, we did not find a similar predictive effect of VIN-like expression in other obesity-associated cancers such as rectal, kidney, and breast adenocarcinoma. To further

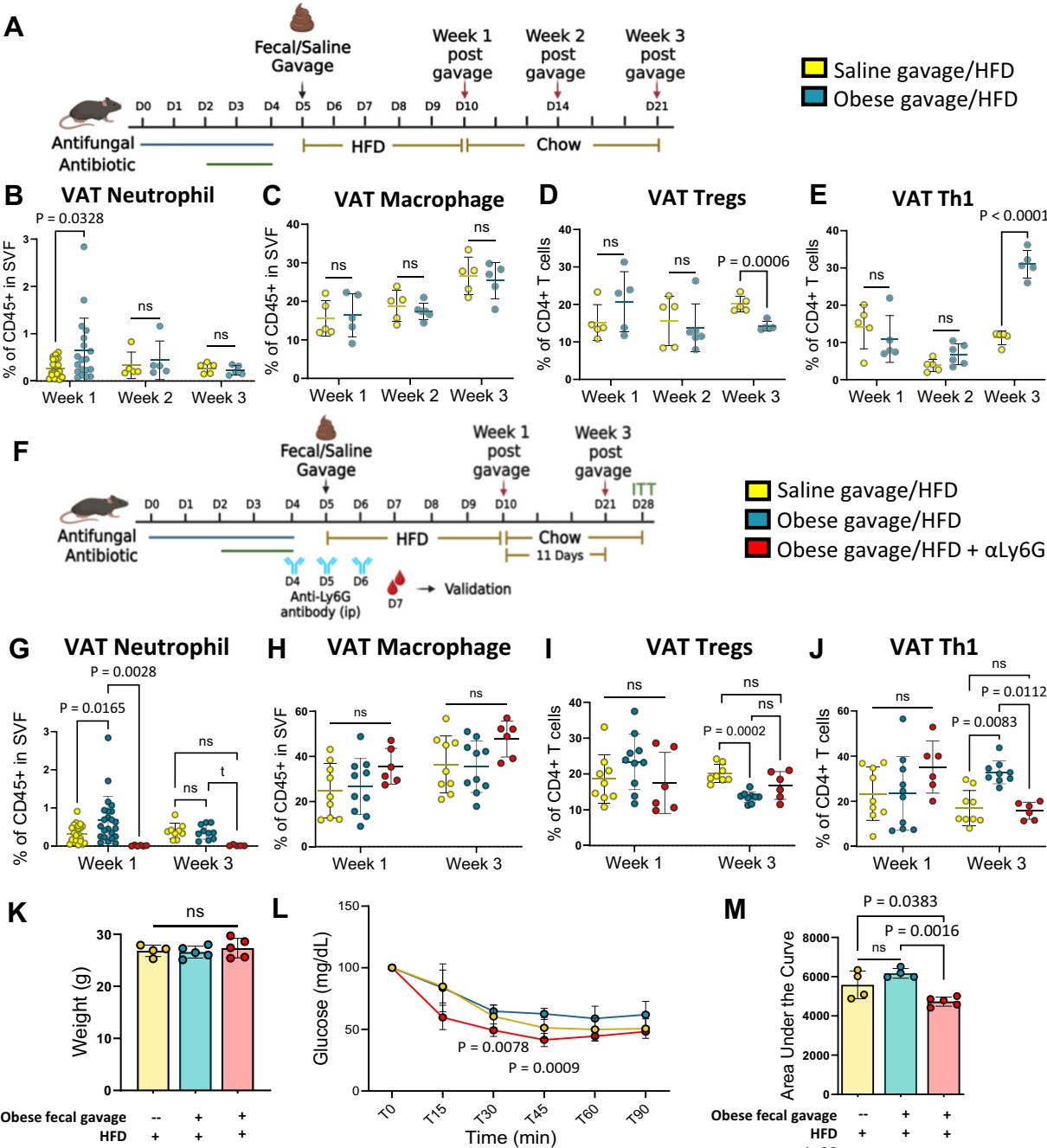

**Fig. 4 | Long-term exposure to obese human fecal microbiome alters VAT immune cell profile, and neutrophils mediate these effects. A** Stool from an individual with obesity vs. saline control was gavaged into mice (*n* = 5 mice/ group) and immune cell changes in VAT were monitored for 3 weeks. **B** In Week 1 post obese fecal gavage, there is a neutrophil rise similar to that shown in Fig. 3, which subsides by week 2, **C** no differences in macrophage population between saline vs. obese fecal gavage, **D** Anti-inflammatory regulatory T cells are substantially reduced by week 3, and in sharp contrast, **E** proinflammatory Th1 cells start to increase by week 2 and are greatly increased by week 3. **F** Administration of neutrophil depletion antibody against Ly6G (αLy6G) for 3 days (*n* = 6 mice/group), **G** eliminated the initial spike of neutrophils in week 1, **H** VAT macrophages remained unaffected, **I** stopped the subsequent drop of Tregs, and **J** stopped the rise of Th1 cells that was observed previously in the presence of neutrophils. A

group of mice (*n* = 5 mice/group) was continued on a chow diet for 1 more week, mice that received neutrophil depletion antibody had **K** unchanged body weights, however, **L** remained insulin sensitive compared to the ones that had an initial spike of VAT neutrophils. Insulin sensitivity was measured by insulin tolerance test (ITT) for 90 min post glucose injections. **M** Area under the curve (AUC) for ITT. All data represented as mean ± SD compared by two-way ANOVA followed by post hoc Tukey's multiple comparisons test with two-tailed analysis. $^{t}p < 0.1$ $^{*}p < 0.05$, $^{**}p < 0.01$, $^{***}p < 0.001$, $^{****}p < 0.0001$. ITT data (*n* = 5 mice/ group) represented as mean ± SD. Statistical analysis was performed using two-way ANOVA and Tukey's post hoc test with two-tailed analysis (glucose levels) or Student's two-tailed *t*-test (AUC). (**A** and **F** created with BioRender.com released under a Creative Commons Attribution-NonCommercial-NoDerivs 4.0 International license https:// creativecommons.org/licenses/by-nc-nd/4.0/deed.en).

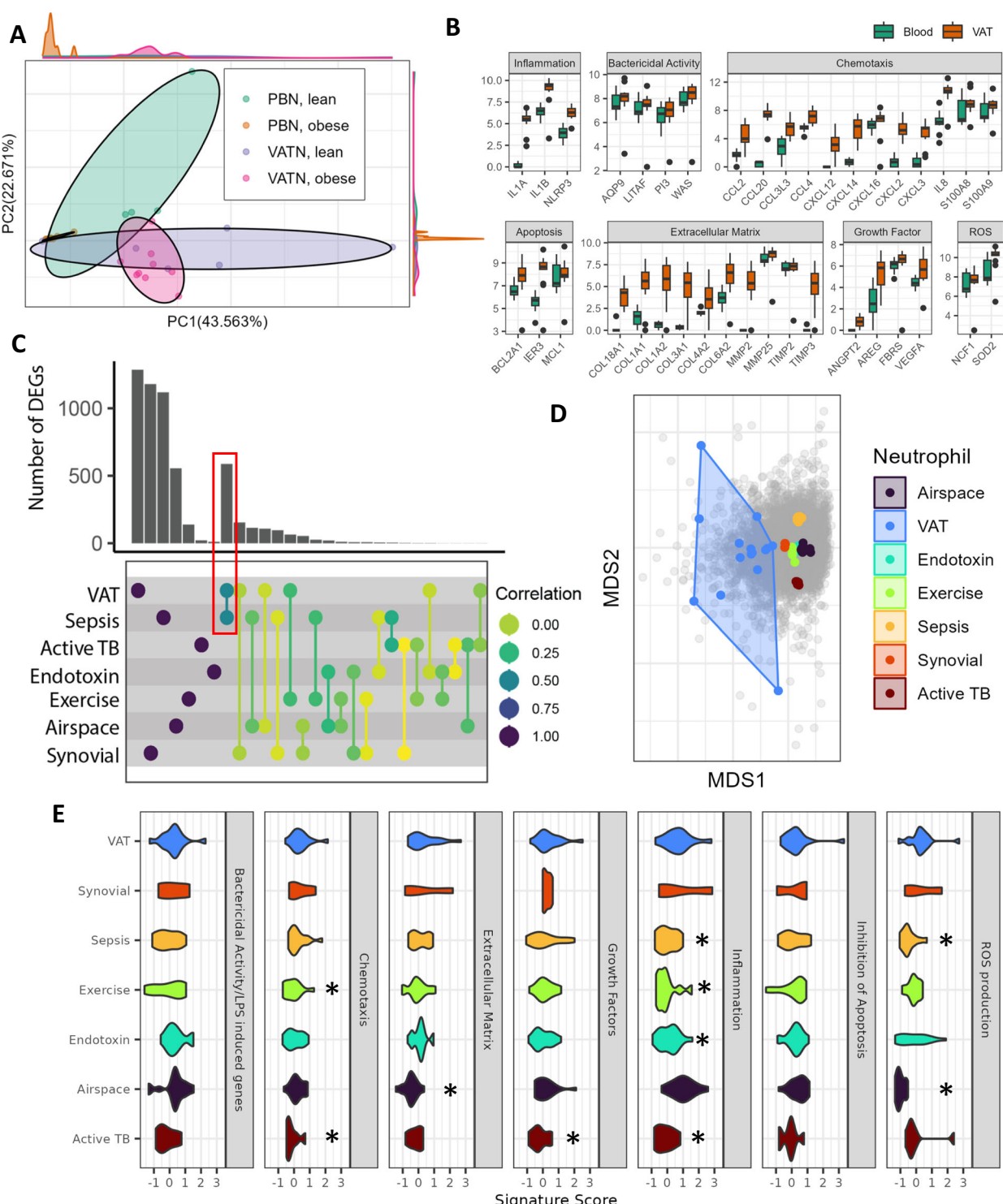

validate the presence of these neutrophils, we gathered publicly available single-cell RNAseq data from lung tumors and assessed the VIN-like expression of the neutrophil clusters. We observed strong VIN-like expression in only a subset of the neutrophils, suggesting similar behavior as observed in the bulk RNAseq data. Since not all of the neutrophils showed the VIN-like phenotype, it likely is not a general characteristic of tissue infiltration (Fig. 6G).

## Discussion

Little is known about the presence and function of VAT neutrophils in human obesity. We found that VAT neutrophils are more abundant in subjects with obesity compared to lean subjects and express more

inflammation and activation-related genes compared to peripheral blood (PB) neutrophils. VAT neutrophils as a % SVF correlated with both circulating levels of LPS and insulin resistance, suggesting VAT neutrophil abundance is related to the presence of bacteria and influences systemic metabolism. Indeed, we identified bacteria within human VAT (collected and processed under sterile conditions), which differed between lean individuals vs. individuals with obesity. A mouse model treated with antibiotics and an antifungal and gavaged with human fecal microbiome showed bacterial translocation to VAT only in the context of an HFD in the recipient mice, while VAT neutrophil abundance was dependent on obesity in the donors, such that mouse VAT neutrophil abundance correlated with BMI of the human donor.

**Fig. 5 | Transcriptomic analyses of neutrophils isolated from VAT are distinct from those isolated from other neutrophil activation states. A** Principal component analysis of gene expression from isolated neutrophils. ($n = 9$ obese, $n = 6$ lean). **B** Expression counts of genes in neutrophil activation pathways are higher in neutrophils isolated from VAT than those isolated from PB. ($n = 9$ obese, $n = 6$ lean). **C** The number of DEGs in each condition, DEGs shared between different neutrophil activation types, and Spearman correlation of DEG expression between conditions. In each group, DEGs are calculated using the isolated neutrophil of interest (e.g., VAT) compared to PB controls from the corresponding data. Vertical bars indicate how many DEGs are shared between experiments, with comparison groups indicated by connected circles. Circle colors indicate the global expression correlation for the DEGs in the experiments. The red box highlights that VAT shares the most DEGs with Sepsis and expression of these genes correlates most strongly. (VAT: $n = 15$ blood, $n = 15$ VAT. Endotoxin: $n = 17$ LPS stimulated, $n = 14$ control. Exercise: $n = 12$ post-exercise, $n = 12$ control. Sepsis: $n = 15$ patients, $n = 8$ control. Active-TB: $n = 7$ patients, $n = 4$ control. Synovial: $n = 6$ synovial, n = 24 blood. Airspace: $n = 17$ airspace, $n = 31$ blood.) **D** Multi-dimensional scaling plot of isolated

neutrophil expression datasets whose DEGs were examined in (**C**). Colored points and hulls indicate samples and circumscription of their clusters, and gray points indicate genes. **E** Signature scores for common neutrophil pathways, comparing VAT-isolated neutrophils to publicly available data of isolated neutrophils. Data are shown with violin plots of each neutrophil type's score densities. For each dataset, signature scores were calculated against their own controls. Neutrophil types with scores significantly different from VAT by the two-sided Kruskal–Wallis test in each pathway are marked with an asterisk. The sample set is identical to (**C**). (Starred $p$-values: Exercise, chemotaxis = 0.0318, Exercise, inflammation = 0.0128, Sepsis, inflammation = 0.0191, Sepsis, ROS = 0.00264, Endotoxin, apoptosis = 0.00891, Airspace, extracellular matrix = 0.00142, Airspace, ROS = 0.0000129, Active-TB, chemotaxis = 0.0316, Active-TB, growth factors = 0.0445, Active-TB, inflammation = 0.0316) Boxplots in (**B**) are defined using first and third quartiles to bound the box. Minima is first quartile $- 1.5 * IQR$, and maxima is third quartile $+ 1.5 * IQR$. Outliers are shown as points. Individual points are not shown due to space constraints.

The increase in VAT neutrophils was followed by an increase in proinflammatory CD4+ Th1 cells and a drop in anti-inflammatory Tregs. However, the T cell changes did not occur if the neutrophil spike was eliminated with a neutrophil-depleting antibody, strongly suggesting neutrophils instigate VAT inflammation. In addition, although VAT neutrophil activation was distinct from other states of neutrophil activation, the use of a custom-signature tool found VAT-type neutrophil expression in other tissues, including adenocarcinoma of the colon, where the VAT-type neutrophil abundance correlated with overall survival. Thus, VAT-type neutrophils may be important not only to metabolic disease but obesity-related cancer progression.

Increasing evidence suggests that alterations in gut permeability and the gut microbiome with HFD contribute to the entry of LPS and gut bacteria into the circulation in humans and mice[40–43]. In fact, when we administered HFD for only two weeks to healthy lean individuals, 9 out of 11 increased their plasma zonulin levels, suggesting increased gut translocation. Recently, the presence of neutrophils within the lumen of AT vasculature in human obesity was reported[44], similar to what occurs in the liver and spleen, while another study demonstrated neutrophil infiltration through the vessel wall[45]. However, the mechanism of the VAT neutrophil accumulation and its importance is unknown. In our investigation, multiple lines of evidence suggest that the translocation of bacteria from the gastrointestinal tract in obesity contributes to VAT neutrophilia: (1) Subjects with obesity have increased plasma levels of LPS binding protein and zonulin; (2) VAT neutrophils correlate with circulating LPS, (3) Ampliseq analyses indicate LPS responsive genes are increased in VAT vs. PB neutrophils including *LITAF* (lipopolysaccharide-induced TNF factor), and TLR2, TLR4, and CD14 which bind to bacterial proteins[46–48] and (4) the bacteria identified in human VAT were most likely gut-derived. IL8 is a powerful neutrophil chemoattractant; human adipocytes from individuals with obesity compared to lean individuals express 10-fold higher levels of IL8, while LPS stimulates IL8 gene expression by nearly 150-fold in cultured adipocytes. Thus, adipocyte IL8, in response to bacteria, likely plays a major role in attracting neutrophils into VAT.

The finding of bacteria in AT has previously been reported in subjects with obesity but without assessment in lean individuals[23]. In that investigation, bacterial load was greatest in mesenteric fat > subcutaneous fat > omental fat and was also detected by fluorescence in situ hybridization of these tissues. In a depot-specific manner, bacterial load correlated with the expression of inflammatory markers and with macrophage infiltration in omental fat but not with BMI, glucose, or lipid measures. Proteobacteria and Firmicutes were the predominant phyla of AT bacteria identified. In our investigation, AT bacteria were similar in lean subjects vs. subjects with obesity and largely comprised of the phyla Firmicutes and Actinobacteria, which are generally gut-derived taxa. Our finding that *Streptococcaceae* and

an unclassified genus in the family *Ruminococaceae* were enriched in VAT from subjects with obesity is consistent with previous reports suggesting *Streptococcaceae* are more common in the gut microbiome of individuals with obesity vs. lean individuals[49], while *Ruminococcaceae* are increased in the intestinal microbiomes of patients with type 2 diabetes compared to those without[50]. *Marvinobryantia*, which consumes oligosaccharides and increases gut levels of succinate, was enriched in lean VAT and suggested to be useful in treating metabolic diseases by increasing energy consumption[51,52]. The order Bacilliales, also enriched in lean AT, has been associated with the alleviation of hyperglycemia in animal models of T2D[53]. These data are, thus, consistent with the gut as an AT microbial source and with differences in gut microbiome between lean individuals and individuals with obesity.

To further provide a causal link between alterations in the gut microbiota and VAT neutrophilia, we gavaged human feces from subjects with obesity and lean subjects into microbiome-depleted mice. Mice gavaged with stool from donors with obesity showed increased levels of VAT neutrophils only if the mouse was ingesting HFD, consistent with our and several previous observations that HFD increases intestinal epithelial transport[41–43]. Interestingly, the increase in neutrophils correlated with the BMI of the donor (Supplementary Fig. 8). No differences were observed in VAT macrophages, suggesting that human-microbiome administration targeted VAT neutrophils. The primary bacteria in VAT of mice gavaged with feces from two donors with obesity was *Pseudomonas*, while *Staphylococcus* was in highest abundance in VAT of mice gavaged with feces from lean donors. *Pseudomonas* is known to attract neutrophils[54], and a landmark study that investigated bacteria in liver and fat (omental, mesenteric, and subcutaneous) deposits of individuals with BMI ≥ 50 kg/m² reported that Proteobacteria was the dominant phylum and *Pseudomonas* was the predominant genus found in all tissues[55], highly consistent with our results. Of note, these investigators identified a number of other bacteria, some from the gut and some from environmental sources like food, soil, and water. Based on these observations, it is not surprising that VAT also shared the same bacteria not only with gut but also with the lung in our study. Lean patients were not studied in that report, but we found that lean fecal microbiomes led to *Staphylococcus* in mouse VAT, which has been shown to suppress LPS-induced inflammation[56]. Further studies are needed to determine the bacterial-specific effects on neutrophil accumulation and inflammation, but these results suggest the type of bacteria plays a role.

Since neutrophils in humans are the most abundant immune cell, contributing to nearly 60% of the circulating immune cells[22], we wanted to ensure that the increase in AT neutrophils in human obesity was not simply due to an increase in migrating PB neutrophils[57]. Ampliseq analyses indicated that AT neutrophils are functionally distinct from circulating neutrophils. These differences include heightened AT

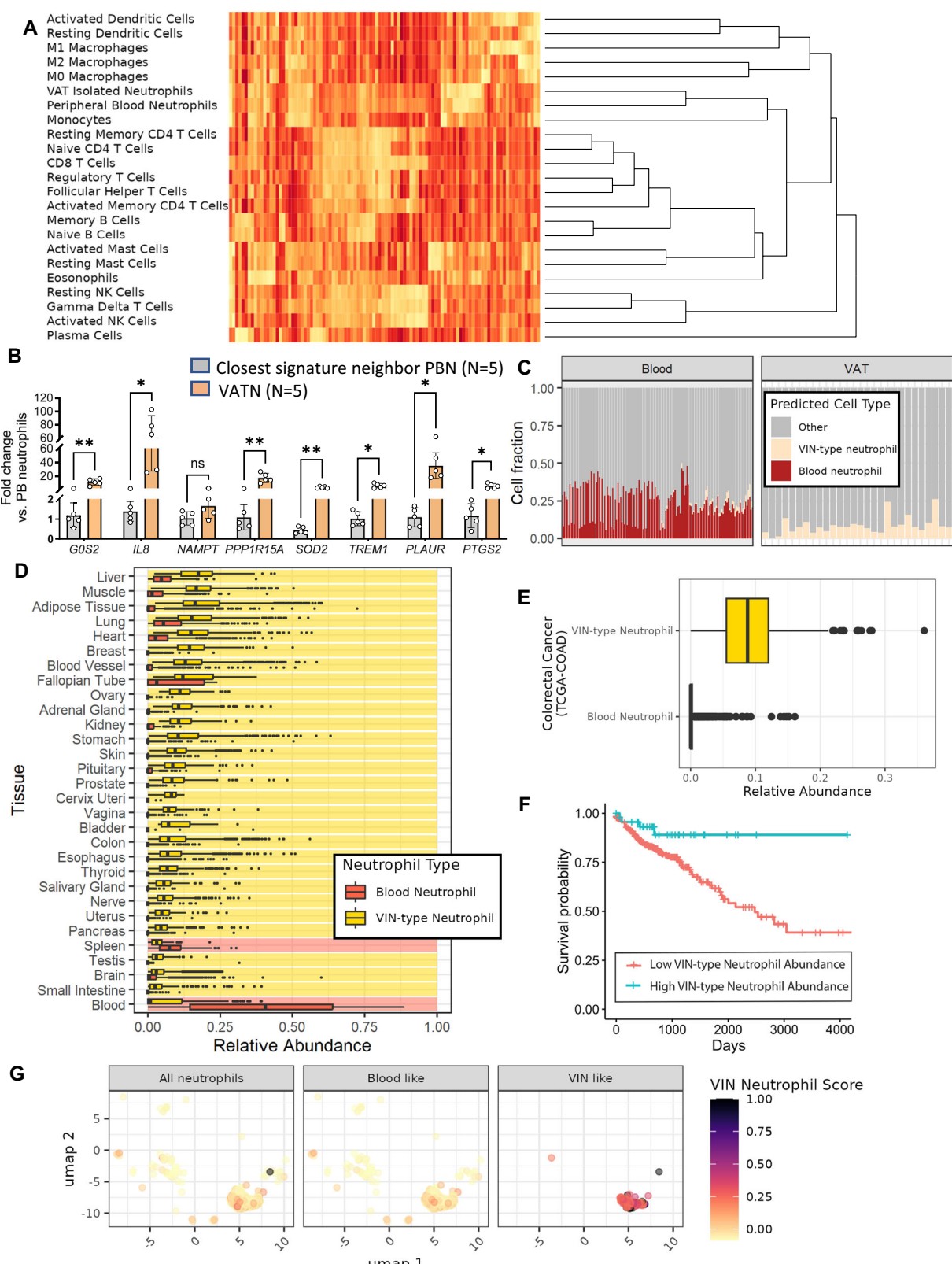

expression of several important chemokines (including *CCL20, CCL2, IL8,* and *S100A9*), which recruit more neutrophils and promote infiltration of T cells and macrophages[58–62]. AT neutrophils also reveal a marked increase in the expression of extracellular matrix (ECM) and other proteins that influence the surrounding microenvironment, including growth factors (vascular endothelial growth factor A [*VEGFA*], fibrosin [*FBRS*], angiopoietin 2 [*ANGPT2*] and amphiregulin

[*AREG*]), extracellular-matrix components including various collagens (*COL6A2, COL1A1, COL1A2, COL3A1, COL18A1,* and *COL4A2*), matrix metalloproteinases (*MMP2* and *25*) and tissue inhibitors of metalloproteinases (*TIMP2* and *3*). Fibrosis is common during AT expansion and impacts adipocyte growth, hypoxia, inflammation, and systemic metabolism. In addition, VAT neutrophils show increased expression of genes related to ROS production and degranulation (neutrophil

**Fig. 6 | A VAT neutrophil expression signature identifies VAT-like neutrophils in diverse tissues and associates with survival in colon cancer. A** Heatmap of the gene expression counts for the deconvolution signature gene set, with hierarchical clustering by expression to display the similarity (dendrogram). **B** qRT-PCR validation of the genes that most strongly separate VAT from peripheral blood neutrophils (VAT; $n = 5$, PB; n = 5), **C** Deconvolution results of publicly available RNAseq datasets from blood or VAT to validate the performance of the custom signature. ($n = 144$ blood, $n = 28$ VAT). **D** Deconvolved abundances of blood and VIN-type neutrophils in publicly available datasets of healthy tissues. Color ribbons indicate whether VIN (yellow) or blood (red) neutrophils were more abundant. ($n = 929$ Blood, $n = 187$ Small Intestine, $n = 2642$ Brain, $n = 361$ Testis, $n = 241$ Spleen, $n = 328$ Pancreas, $n = 142$ Uterus, $n = 619$ Nerve, $n = 162$ Salivary Gland, $n = 653$ Thyroid, $n = 1445$ Esophagus, $n = 779$ Colon, $n = 21$ Bladder, $n = 156$ Vagina, $n = 19$ Cervix Uteri, $n = 245$ Prostate, $n = 283$ Pituitary, $n = 1809$ Skin, $n = 359$ Stomach, $n = 89$

Kidney, $n = 258$ Adrenal Gland, $n = 180$ Ovary, $n = 9$ Fallopian Tube, $n = 1335$ Blood Vessel, $n = 459$ Breast, $n = 861$ Heart, $n = 578$ Lung, $n = 1204$ Adipose Tissue, $n = 803$ Muscle, $n = 226$ Liver). **E** Deconvolved abundances of blood and VIN-type neutrophils in publicly available colon adenocarcinoma RNAseq samples. ($n = 478$). **F** Kaplan–Meier curve of publicly available colon adenocarcinoma RNAseq samples stratified by low vs. high VIN-like neutrophil abundance. ($n = 430$ low VAT-like, 48 high VAT-like). **G** Publicly available scRNAseq of immune cells from lung tumors show that the VIN-type neutrophil signature is strongly expressed in only a subset of the total neutrophils. ($n = 153$ VIN-like neutrophils, $n = 7145$ blood-like neutrophils). Gene expression data in (**B**) represented as mean ± SD compared by using paired $t$-test with two-tailed analysis $*p < 0.05$, $**p < 0.01$. The boxplots in (**D**) and (**E**) are defined using the first and third quartiles to bound the box. The minima are the first quartile less 1.5(inter-quartile range) and the maxima are the third quartile plus 1.5(inter-quartile range). Outliers are shown as points.

cytosolic factor 1 [*NCF1*], cytochrome B-245 alpha chain [*CYBA*], superoxide dismutase 2 [*SOD2*]), suggesting increased proinflammatory function of VAT neutrophils. Finally, levels of the antiapoptotic proteins (MCL1 apoptosis regulator, BCL2 family member [*MCL1*], immediate early response 3 [*IER3*], and BCL2 related protein A1 [*BCL2A1*]) are elevated in VAT neutrophils, allowing them to persist in VAT. Taken together, these data suggest that AT neutrophils have a unique genetic signature that is distinct from PB neutrophils.

Differences between human VAT neutrophils from lean individuals vs. individuals with obesity were also identified but were far fewer than between PB and VAT. Pathways with the most differentially expressed genes included the olfactory system and the PI3kinase pathway, both enriched in lean VAT neutrophils. Multiple olfactory receptors are G-protein coupled receptors (GPRs) that mediate chemosensory function. GPRs are abundant in neutrophils and can be activated by short-chain fatty acids (SCFA); for example, human GPR43 mediates the effect of SCFA on neutrophil chemotaxis and calcium mobilization[63]. SCFA are abundant in the AT microenvironment and are also the end products of fermentation by gut bacteria, and from the gut, SCFAs are readily absorbed into the circulation[63–66]. The PI3K pathway mediates insulin action, which has been shown to enhance chemotaxis, phagocytic, and bactericidal activities of neutrophils[67]. A decrease in genes involved in the PI3K pathway may contribute to AT neutrophil insulin resistance in obesity, which may contribute to impaired immunity against infection[68].

We then compared the activated VAT neutrophils with other activated human neutrophils isolated from blood and tissues where transcriptomic data were available. VAT neutrophils are positively associated with neutrophils activated by endotoxin (LPS), consistent with our ampliseq data, and further suggesting bacteria or their products may contribute to VAT neutrophil activation. In contrast, neutrophils predominant during exercise and tuberculosis infection were negatively correlated. We further used a CIBERSORT approach to identify VAT neutrophils using their expression profile compared to other immune cells in tissues containing a mixture of cell types[39], specifically identifying genes differentiated in VAT neutrophils from its closest neighbor in this analysis, PB neutrophils. Validated genes were involved in inflammation and metabolism. *GOS2*, which is shown to negatively regulate lipolysis by inhibiting the ATGL TAG hydrolase activity in AT both in vitro and in vivo, localizes in lipid droplets and prevents their degradation by ATGL[69]. *GOS2* also promotes apoptosis by encoding for a mitochondrial protein that inhibits the anti-apoptotic regulator BCL2[70]. *PTGS2* promotes inflammation by participating in the synthesis of prostacyclin[71]. *IL8*, a neutrophil activation and recruitment marker, is highly expressed in VAT neutrophils[72]. *PLAUR*, also known as *UPAR*, aggregates in neutrophils to initiate activation that increases their proinflammatory potency, leading to superoxide release from polymorphonuclear neutrophils[73], which also explains an increase in *SOD2* expression in VAT neutrophils. *TREM1* (Triggering

Receptor Expressed on Myeloid cells) is activated by various proinflammatory cytokines in the acute phase of inflammation[74]. *NAMPT* is a neutrophil maintenance and maturation gene[75], while *PPP1R15A* is related to an unfolded protein response and ER stress[76]. Overall, validation of these signature genes suggests that neutrophils reside in a more challenging environment in VAT and express genes to promote inflammation and adapt to a lipid-rich environment.

These genes were used to create a VIN-type signature tool to test whether the VIN-type neutrophil and its proinflammatory properties may be common in various tissues. We used a support vector regression framework to find the most distinguishing VIN-type neutrophil genes among 22 other publicly available, high-quality reference datasets of isolated immune cell expression. This tool enabled a search for the VIN-type signature across diverse healthy and diseased tissues by applying a deconvolution framework to bulk-RNAseq data. After validating the performance of the deconvolution signature tool against publicly available datasets and independent biological replicates, we applied this signature tool broadly, in the end resulting in a survey of roughly 18,000 RNAseq samples from over 40 tissues. Surprisingly, the VIN-type signature was significantly more common than blood-type neutrophils in nearly all tissues except the blood-laden spleen and blood itself. This result suggests that neutrophils showing VIN-type expression and their proinflammatory mechanisms may be widespread across the human body and agrees with recently published data showing that neutrophils infiltrate normal tissues to maintain homeostasis, but can also instigate pathology[77]. Neutrophils were not only found in the bone marrow and spleen, but in the lungs, intestine, liver, muscle, skin, and white AT of normal mice, and in the lung, their presence was associated with tumorigenesis. Neutrophil plasticity is known, and like other immune cells, they adapt to their environment, particularly playing dual roles in cancer[78]. Whether bacteria are normally present in these tissues should be pursued.

In light of the abundance of the VIN-type signature in many tissues and the association between obesity and cancer, we applied the signature tool to publicly available expression data from obesity-associated cancer. Colon adenocarcinoma RNAseq showed significantly more neutrophils resembling VINs than PB-type neutrophils, consistent with the survey of healthy tissues. This cancer occurs more frequently in obesity[79,80]. Further, the survival of patients with colorectal cancer correlated with high versus low stratification by VIN-type expression, whereas no such association with survival could be observed with the abundance of PB-type neutrophils. Thus, neutrophils resembling VINs indeed play an important role in health in diverse contexts and warrant further exploration, especially in the context of cancer. Neutrophils are known to affect tumor initiation and proliferation via involvement in tissue regeneration following injury, which can establish a pro-tumor inflammatory microenvironment[78]. Notably, several recent reports have described the presence of

bacteria in tumor tissue, which may suggest a mechanism for the recruitment of neutrophils resembling VIN[81,82].

## Limitations
Avatar mice appear to be a useful model to examine conditions and mechanisms of how human fecal microbiome affects VAT immune cell composition, although mice, compared to humans, have a short-lived VAT neutrophil spike, compared to the chronic elevation seen in humans with obesity. It is possible that repeated exposure to the human gut microbiome in subjects with obesity may lead to more chronic VAT neutrophilia in mice. Nevertheless, the model has allowed us to investigate early processes by which HFD and gut bacteria lead to VAT inflammation. Although HFD increases gut bacterial translocation, other sources of bacteria that are found in tissues exist, as suggested by Anhe et al.[55], so that may explain why we found that not only gut and VAT share bacteria but lung and VAT also share bacteria. Although the loss of neutrophils caused by the αLy6G antibody substantially reduced insulin resistance in mice gavaged with human fecal microbiome from subjects with obesity, we did not detect a difference in insulin sensitivity in mice who had received saline vs. obese fecal gavage, which might be anticipated since the decrease in neutrophils with antibody administration was more profound than either of these groups. Moreover, none of these mice were obese, so more weight changes may have been necessary to detect differences in insulin responsiveness in these groups of mice.

In summary, these results provide evidence that neutrophils are important, potentially early, contributors to the chronic, low-grade inflammation seen in VAT from subjects with obesity and associated with both insulin resistance and cancer survival. Therefore, therapies targeting alterations in gut leakiness and/or early infiltration of neutrophils into AT and other tissues, like cancer, may provide insight into novel targets for managing AT inflammation and developing the inflammatory complications associated with obesity.

## Methods
All human and mouse studies were approved by The Ohio State University Institutional Review Board (IRB # 2014H0471) and The Institutional Animal Care and Use Committee (IACUC# 2014A00000108-R3), respectively.

### Human subjects study design
Eligible patients were recruited from The Center of Minimally Invasive Surgery at The Ohio State University in Columbus, OH. Patients were excluded from the study if they were taking a chronic steroid or anti-inflammatory agent, had end-stage renal or liver disease, or had a past diagnosis of Acquired Immune Deficiency Syndrome (AIDS) or neoplastic disease. A total of ninety-six patients were recruited (ages 21–75, BMI 18–50 kg/m$^2$). Lean patients were scheduled to undergo elective, non-emergent intra-abdominal surgery (primarily cholecystectomies and hernia repairs). In contrast, patients with obesity were scheduled to undergo either Roux-en-Y gastric bypass or sleeve gastrectomy bariatric surgery. All patients gave informed consent and underwent medical evaluation before study enrollment.

### Biospecimen collection
Blood samples were obtained during pre-operative care and before the administration of anesthesia. The blood was placed on ice until further processing within the laboratory. Visceral adipose tissue (VAT) biopsies were obtained at the time of surgery before gut manipulation in both patients with obesity and lean patients. The samples were processed within thirty minutes and separated into adipocytes and the stromal vascular fraction (SVF) as detailed below in "Adipose tissue processing" section[38]. While low microbial biomass samples are extremely prone to contamination from laboratory reagents[83], instruments[84], and environment[85], we went to great lengths to avoid

these issues within our 16S DNA-based analysis. During sample acquisition, AT was collected in a sterile surgical field and immediately transferred to sterile containers filled with sterile phosphate-buffered saline. Samples were taken directly to the laboratory, where they were opened under the hood, and a small piece of AT was removed using sterile scissors before flash freezing and transferred to the -80 °C freezer for further microbial analyses.

### Adipose tissue processing
Fresh omental adipose tissue (AT) (5–20 g) was processed within 30 min of procurement. 1/2 g was flash-frozen, and maintained sterile at −80 °C until use for 16S RNA metagenomics. Briefly, adipose tissue was minced into 2–3 mm pieces and digested with collagenase buffer (Gey's balanced salt solution, Sigma G9779; 1 M HEPES, Gibco 15630-080; 10% BSA, Fisher bioreagents BP1605-100; 5 mM adenosine, Thermo Scientific A10781.09; and collagenase 2, Worthington Biochemical LS004176) for 45 min at 37 °C. Neutralization of digestion included DMEM with 3% FBS (MilliporeSigma F2442) and 1% P/S (Sigma P4333). Digested AT was filtered through 500 μm and 300 μm filters, respectively, followed by centrifugation at 1100 rpm for 10 min at 4 °C. The top layer of adipocytes and bottom pellet of stromal vascular fraction (SVF) were collected for RNA extraction and FACS sorting or flow cytometry analysis, respectively. Fractionated SVF was prepared for flow analyses post-RBC lysis (Gibco A10492-01). Details are available in our previous publications for mouse and human VAT collection[17,86]. We employ sterile collection and processing procedures for all tissues, every sample is processed individually with autoclaved instruments for mincing the tissue. Buffers are made fresh for each human sample or mouse experiment and are not reused if opened outside of the culture hood. The flow cytometer is cleaned and calibrated after each use to avoid contamination.

### Neutrophil isolation
For peripheral blood isolation of neutrophils, Ficoll gradient and dextran sedimentation were used as previously described[87]. Purified cells were spun down and resuspended in RNA lysis buffer for RNA extraction using the Quick-RNA Isolation kit with in-column DNase treatment (Zymo Research, Irvine, CA). VAT neutrophils were stained for FACS sorting using a standard flow staining protocol. Neutrophils were sorted by flow cytometry (BD FACSAria™ III Cell Sorter) as LIN-CD3 (Cat# 300440, Clone: UCHT1), CD14 (Cat# 325604, Clone: HCD14), CD19 (Cat#302206, Clone: HIB19), CD20 (Cat# 302304, Clone: 2H7), CD56 (Cat# 318304, Clone: HCD56), CD15+ (Cat# 301906, Clone: HI98), CD11b+ (Cat# 301310, Clone: ICRF44), and CD66b+ (Cat# 305108, Clone: G10F5). Cells were collected into media with 10% FBS for cell recovery. Isolated neutrophils were subjected to Ampliseq analyses as previously described[88].

### Histology
The whole fat was fixed for 48 h in 10% formalin, transferred to 70% ethanol, and stored at 4 °C. Samples were paraffin-embedded, and sequential slides were cut to a thickness of 10 μ for H&E staining at the histology core at The Ohio State University. A pathologist reviewed slides of 6 subjects with obesity and 4 lean subjects for neutrophil identification. Bright-field images were captured using a light microscope (Zeiss Axio Observer upright microscope).

### Human blood and VAT flow cytometry and sorting
For human samples, neutrophils were defined as LIN- (FITC: CD3 (Cat#:300440, Clone: UCHT1), CD14 (Cat#: 325604, Clone: HCD14), CD19 (Cat#: 302206, Clone: HIB19), CD20 (Cat#: 302304, Clone: 2H7), CD56 (Cat#: 318304, Clone: HCD56)), CD16 (APC, Cat#: 302012, Clone: 3G8), CD11b (BV421, Cat#: 301324, Clone: ICRF44), and CD15 (BV605, Cat#: 323032, Clone: W6D3). They were also further characterized by markers of activation and migration: CD66b (PerCP-Cy5.5, Cat#:

305108, Clone: G10F5) and CD62L (PE, Cat#: DREG-56, Clone: 304805), respectively. All antibodies were purchased from BioLegend unless otherwise specified. Fixable Viability Dye eFluor 506 (eBioscience, Cat#: 65-0866-14) was used to exclude dead cells from analysis. Flow cytometer, BD LSRII was used to run the stained samples through the Flow Cytometry Core in the OSU Comprehensive Cancer Center. All samples were analyzed with FlowJo software. The representative gating strategy is demonstrated in Supplementary Fig. 1.

## Circulating plasma markers

Commercially available ELISA kits were purchased from EMD Millipore and completed as per the manufacturer's protocol for the measurement of circulating insulin, adiponectin, and leptin (Cat#s: EZHI-14K, EZHADP-61K, and EZHL-80SK, respectively). ELISA kits were also purchased from Cusabio for lipopolysaccharide (LPS, Cat#: CSB-E09945h) and serum zonulin (Cat#: CSB-EQ027649HU) and from R&D Systems for LPS binding protein (LBP, Cat#: DY870-05). Glucose was measured using the Glucose Liquicolor (Stanbio Laboratory, REF#1070-125) regent kit according to the manufacturer's protocol.

## Bacterial Amplicon (16S) sequencing and analysis

The bacterial 16S rRNA gene was amplified from VAT or stool samples from humans with and without obesity or from mice. VAT was surgically collected as described above and immediately flash-frozen. Samples were thawed and centrifuged at 11,200 RCF for five minutes at room temperature to separate into layers. The interphase layer (-100 μL) was collected from each sample and added to a tube containing lysis buffer and 200 μL garnet beads. Cells were lysed on a PowerLyzer 24 at 2000 rpm for 30 s, and then DNA was purified using an AllPrep kit (QIAGEN). The bacterial rRNA was amplified using V3-V4 primers and KAPA HiFi enzyme (50 C 30 s, 72 C 2 × 20 cycles). Amplicons were cleaned by magnetic beads, and then sequencing libraries were generated using a QIAseq kit (QIAGEN) following the manufacturer's instructions. Libraries were sequenced on a MiSeq 2 × 250 using a V3 reagent kit (Illumina). Demultiplexed fastqs were filtered for quality and length (340–440 bp) and then processed to amplicon sequence variants (ASVs) using "dada2" version 1.12.1 in R[89]. Taxonomy was assigned by comparison to the SILVA database[90]. Relative abundances of taxa were used for analysis.

All bacterial isolation and amplification steps occurred in a HEPA-filtered PCR cabinet (AC600, AirClean Systems) after a thorough cleaning and 30 min UV treatment. Lysis buffer blanks were processed through each cell lysis step, nucleic acid purification, amplification, and library preparation protocols and then sequenced. Following data processing, sample contaminants were first statistically filtered using the "decontam" package in R[91]. Genera of any ASVs observed in the negative controls were removed from the samples. In cases where the negative control ASVs were not identified to the genus level, sample ASVs 97% similar to negative controls were filtered.

The number of genome copies per microliter DNA sample was calculated by dividing the gene copy number by an assumed number of gene copies per genome (four). The amount of DNA per microliter DNA sample was calculated using an assumed genome size of $4.64 \times 10^6$ bp, the genome size of Escherichia coli. The sources of the microbes found in VAT samples were estimated using sourcetracker2[92]. Source locations included all body sites from the Human Microbiome Project[93] (all subsites of the airway, gastrointestinal tract, urogenital tract, oral, and skin) and the MGnify "built environment" biome[94]. Data were retrieved using the HMP16SData[95] and MGnifyR packages[94].

## Human weight gain protocol

As a part of a previously conducted and published study[17], a separate group of lean metabolically healthy patients ($n = 11$, 9 males and 2 females, age $27.0 \pm 7.0$ yo, baseline BMI $22.4 \pm 1.8$ kg/m²) were

consented to consume at minimum an additional 1320 kcal/day with >50% of total caloric intake in total fat and >10% in saturated fats by dining at fast-food restaurants. Subjects were instructed not to change their level of physical activity or start any new medications during the study. Pre and post-HFD plasma samples were collected, and ELISA was performed to measure plasma zonulin levels (Human Zonulin ELISA kit, Elabscience, Cat# E-EL-H5560). The study was approved by The OSU institutional review board, and all participants provided written informed consent. All subjects were compensated for participation.

## Human fecal microbiome collection

Fecal material from 4 individuals with obesity (BMI 30.5–43.6) and 3 lean (BMI 18.8–21.9) was purchased from Medix Biochemica Group. The demographics of the donors are listed in Supplementary Table 1. The fecal material was kept sterile and frozen from the time of collection until use. Exclusion criteria for fecal material donors: Any major illnesses or chronic disease, taking chronic medications (birth control allowed), taking medications that could impact the immune system, taking antibiotics, antifungal or nonsteroidal, smoking, taking excess alcohol or social drugs.

## Mouse experiments

Eight-week-old C57BL/6 male mice from Jackson lab (strain 000664) were used for this study. Upon arrival at the OSU animal facility, the mice were rested for a week before starting the experimental protocol. All animals were housed in groups of five mice in standard rectangular cages. They were treated with a broad-spectrum antibiotic cocktail (MilliporeSigma-A5354, N1142, 627850, G1272) and an antifungal drug (MilliporeSigma A2942) for four days to deplete the resident mouse microbiome[96]. Fecal pellets were collected at the end of 4 days to assess the efficiency of microbiome depletion. On the next day, human fecal slurry from lean donors and donors with obesity was prepared using saline, and 200 μl was gavaged (oral) into the microbiome-depleted mice ($n = 5$–7 mice/ gavage or diet group). A control group was gavaged with saline. These mice were then kept on a high-fat diet (HFD; 60% kcal% fat; D12492, Research Diets, New Brunswick, NJ) or normal chow for five days. At the end of five days, fecal pellets were collected for the microbiome analysis, and mice were sacrificed. Epididymal visceral adipose tissue (eVAT), spleen, liver, and lung were harvested for flow cytometry, gene expression analysis, and 16S sequencing, the brain was also collected as a control organ for 16S Sequencing. For a long-term experiment, to assess the consequences of the VAT neutrophil increase, mice were switched to normal chow 5 days after the gavage and sacked at days 14 and 21 after the start of HFD/ oral-fecal gavage. Flow analyses were performed in AT SVF.

Using the same experimental protocol as above, a neutrophil depletion experiment was continued in a separate 8-week-old C57BL/6 mouse cohort using inVivo MAb anti-mouse Ly6G antibody (BE0075-1, Bio X cell USA) at 1 g/kg for each mouse. Three intraperitoneal injections were given to the mice on days −1, 1, and 2 of oral-fecal gavage and HFD feeding start day. Post 3 injections, to validate the efficacy of neutrophil depletion, 100–150 μl of blood was collected using the submandibular bleeding method, and flow analysis was done for neutrophils (33111080). Briefly, freshly isolated blood (EDTA tubes) was centrifuged at $400 \times g$ for 7 min at 4 °C. On the pellet, the RBC lysis step was done with ACK lysis buffer (Biolegend,420302) and one PBS wash was given. The pellet was then resuspended in FACS buffer and subjected to flow cytometry.

## Mouse flow cytometry analysis

Mouse adipocyte and adipose stromal vascular fraction (SVF) were isolated from epididymal VAT as previously described above[38]. Spleens were minced and filtered through a 70 μm strainer to get Splenocytes. RBCs were lysed using ACK lysis buffer (Biolegend,420302). The cells were washed and resuspended in FACS buffer. Splenocytes and SVF

cells were Fc blocked and stained with respective fluorochrome-conjugated antibodies directed against required mouse antigens for flow analysis; viability staining was done using viability dyes (Biolegend 423104). Intracellular staining was performed for intracellular transcription factors (Foxp3, Tbet, etc.) Post fixation and permeabilization steps using transcription factor staining buffer set (eBioscience 00-5523-00) following manufacturer's protocol. Cytek Aurora flow cytometer was used to run flow and was analyzed with FlowJo v10.8.1 (Tree Star) software.

For flow cytometric analysis of VAT and splenic neutrophils- (Lin-CD3(Cat#100204, Clone:17A2), CD19(Cat# 115506, Clone:6D5), NK1.1(Cat#108706, Clone:PK136), CD45+(Cat#103126, Clone:30-F11), CD11b+(Cat#101257, Clone:M1/70), Ly6G+(Cat#127648, Clone:1A8), CD64-(Cat#139311, Clone:X54-5/7.1), CD11c-(Cat#117328, Clone:N418)). Macrophages- (Lin- CD3(Cat#100204, Clone:17A2), CD19(Cat#115506, Clone:6D5), NK1.1(Cat#108706, Clone:PK136), CD45+(Cat#103126, Clone:30-F11), CD11b+(Cat#101257, Clone:M1/70), CD64+(Cat#139311, Clone:X54-5/7.1), F4/80+(Cat#123137, Clone:BM8)). Regulatory T cells (Tregs)- (Lin- CD19(Cat#115506, Clone:6D5), Gr-1(Cat#108406, Clone:108406), NK1.1(Cat#108706, Clone:PK136), CD45+(Cat#103126, Clone:30-F11), CD3+(Cat#100204, Clone:17A2) CD4+(Cat#100467, Clone: GK1.5), CD25+(Cat#101910, Clone:), Foxp3+(Invitrogen, Waltham, MA,USA, Cat#12-5773-82, Clone:FJK-16s)). T helper 1 (Th1) cells- (Lin- CD19(Cat#115506, Clone:6D5), Gr-1(Cat#108406, Clone:108406), NK1.1(Cat#108706, Clone:PK136), CD45+(Cat#103126, Clone:30-F11), CD3+(Cat#100204, Clone:17A2) CD4+(Cat#100467, Clone: GK1.5), CD25-(Cat#101910, Clone:), Tbet+(Cat# 644824, Clone:4B10) CXCR3+(Cat#126531, Clone:CXCR3-173)). Fixable Viability dye Zombie yellow (Cat#: 423104) was used to exclude dead cells from analysis. All antibodies from Biolegend, CA, USA, were used unless otherwise specified. Representative gating strategies are demonstrated in Supplementary Figs. 6 and 7.

For mouse lung and liver, freshly harvested tissues were cut into small pieces, followed by digestion in RPMI media containing collagenase IV (1 mg/ml; ThermoFisher; Cat#17104019) and DNase I (0.25 mg/ml; Millipore sigma; Cat#10104159001) while shaking for 45 min at 37 °C. The digested cell suspension was filtered through a 40 μM cell strainer to obtain a single-cell suspension. The red blood cells were lysed using ACK lysis buffer for 5 min at room temperature. The cells were washed and resuspended in FACS buffer (PBS containing 1% BSA and 2 mM EDTA). For flow cytometry, cells were kept at 4 °C and blocked using anti-CD16/32 (BD, Cat#: 553141, Clone: 2.4G2) before being stained with appropriate primary antibodies for 30 min. For neutrophils- Lin- CD3(Cat#100204, Clone:17A2), CD19(Cat# 115506, Clone:6D5), NK1.1(Cat#108706, Clone:PK136), CD45+ (BV510, Cat#: 103137, Clone: 30-F11), CD11b+ (BV711, Cat#: 101241, Clone: M1/70) and Ly6G+ (APC-Cy7, Cat#: 127623, Clone: 1A8). All antibodies were purchased from BioLegend unless otherwise specified. Fixable Viability dye Zombie UV (Cat#: 423107) was used to exclude dead cells from analysis. Cells were washed and fixed with Cytofix fixation buffer (BD, Cat#: 554655) before being analyzed by BD LSR Fortessa flow cytometry in the OSU Comprehensive Cancer Center. All samples were analyzed with FlowJo v10.8.1 (Tree Star) software.

All flow cytometry data is represented as mean ± SD. Statistical analysis was performed in GraphPad Prism. Comparisons between two groups were performed by the two-tailed unpaired student's $t$-test unless otherwise specified. Comparison of more than two groups was performed using the post hoc Tukey's multiple comparisons test after the analysis of variance (one-way ANOVA) with two-tailed analysis. $*p < 0.05$, $**p < 0.01$, $***p < 0.001$, $****p < 0.0001$.

## Mouse insulin tolerance test (ITT)
ITT was performed using a standard metabolic tolerance test protocol, as previously described in our publications[38]. Briefly, mice were fasted for 6 h prior to the test. ITTs were performed on non-anesthetized

mice, using tail vein blood samples obtained 0, 15, 30, 45, 60, and 90 min after intraperitoneal (IP) insulin injection (0.75U/kg) to obtain the glucose (mg/dL) levels. ($n = 4–5$ animals/ group). All animal procedures were conducted at The Ohio State University in accordance with institutional animal care and use committee guidelines.

## Expression analysis and comparison to other neutrophil activation types
Publicly available data from NCBI's database of GEO datasets were used to acquire samples of peripheral blood neutrophils from individuals in various states of health to compare to neutrophils isolated from visceral adipose tissue. These health conditions included sepsis, exercise, exposure to endotoxin LPS, and active tuberculosis. Expression profiles for samples from whole blood and visceral adipose tissue were also obtained to test our signature matrix. These samples and their accompanying metadata were then manipulated into R using the package "GEOquery." A single probe ID was chosen to represent each gene using the "collapseRows" function.

Differential expression analysis was performed using the R package DESeq2. Unnormalized expression counts were provided to the function, and the p-value was adjusted for multiple hypothesis testing following the method of Benjamini and Hochberg[97]. This process was performed for each of the health states neutrophils were isolated from, and the results were compared to determine how similar VAT neutrophil activation was to other activation states by checking how many significant genes were shared between them.

## VAT neutrophil signature and deconvolution
A custom-signature gene matrix including blood and VAT neutrophils was created using CIBERSORT by substituting the blood and VAT neutrophil data generated here for the existing neutrophil in the LM22 cell signature reference sample set[39]. To test the ability of the signature to identify neutrophil sources, blood and adipose tissue samples were accessed from GEO (Supplementary Table 4) and deconvolved using the custom signature. Clustering analyses used Euclidean distance and the complete linkage method using the stats package in R.

## Gene expression analysis of VAT and PB neutrophils in a validation cohort
Total RNA was extracted from VAT and peripheral blood neutrophils using Zymo Research RNA isolation kit (R2052), followed by a DNase treatment. RNA was then reverse transcribed using the BIOLINE cDNA synthesis kit (BIO65053). BIOLINE SensiFAST SYBR master mix (BIO-98020) was used to perform qRT-qPCR and expression was quantified using ΔΔCt method, normalized to the housekeeping genes *PPIA* (human) or *Ppia* (mouse). All the gene expression data is represented as mean ± SD. Statistical analysis was performed in GraphPad Prism. Comparisons between two groups were performed using a two-tailed student t-test unless otherwise specified. $*p < 0.05$, $**p < 0.01$, $***p < 0.001$. A list of all primers used to perform qRT-qPCR is included in Supplementary Table 6.

In order to compare the impact of different isolation methods VAT and PB neutrophils on signature gene validation, we performed PB neutrophil isolation in 3 different ways prior to RNA isolation for the validation of the neutrophil signature genes ($n = 5$) in- (1) Flow sorting the PB neutrophils, (2) Dextran sedimentation, and 3() to eliminate the effect of VAT specific regents on gene expression, we treated PB neutrophils similarly as VAT neutrophils (with dissociation buffers, temperatures, incubations, etc.) prior to RNA isolation. (Supplementary Fig. 11).

## VAT-like signature in a scRNA dataset
A publicly available dataset of scRNA data was obtained from Prazanowska and Lim[98]. Cells were filtered to those labeled as neutrophils. A simpler VAT-like signature was generated by finding the top ten genes

most different between VAT and peripheral blood in the custom-signature genes and used to generate a VAT-like score for each cell using "tmesig".

## Reporting summary

Further information on research design is available in the Nature Portfolio Reporting Summary linked to this article.

## Data availability

The gene expression and microbiome sequencing data generated for this manuscript are publicly available through the NCBI Bioproject ID: PRJNA766535. Additional publicly available data from NCBI can be accessed at GSE2322, GSE19443, GSE64457, GSE8668, and GSE116899. These accession numbers are further described in Supplementary Tables 2 and 4 with the labels used in our analyses. Cancer expression data were accessed from The Cancer Genome Atlas Project (TCGA) by using the R package "genomicDataCommmons" and applying filters to choose RNASeq samples from colon adenocarcinoma cancers. Additional tissue expression data were accessed from the Genotype-Tissue Expression Project (GTEx). Single-cell RNASeq data with umap values were originally generated by Prazanowska and Lim[98], and can be accessed at https://doi.org/10.6084/m9.figshare.c.6222221.v3. For 16S classification, the SILVA database version 123 [https://www.arb-silva.de/download/archive/] was used. All other data used to generate analyses and visualizations are available within the GitHub repository at https://github.com/spakowiczlab/atbac or using the https://zenodo.org/badge/latestdoi/271302970, including the results of 16S sequencing of mouse samples as provided by Zymo, human ampliseq expression counts, and data prepared for visualizations.

## Code availability

R analyses were run using Rv4.1.0, with some initial processing in Rv4.0.2. R packages used in the access and analysis of data include dada2 v1.12.1, tidyverse v2.0.0, ggdendro v0.1.23, vegan v2.5.7, broom v1.0.4, readr v2.1.4, survival v3.3.1, survminer v0.4.9, tmesig v0.1.0, RColorBrewer v1.1.3, viridis 0.6.5, ggrepel v0.9.1, ggfortify v0.4.14, ggforce v0.3.3, flextable v0.7.2, Seurat v4.4.0, decontam v1.4.0, HMP16SData v1.8.2, MGnifyR v0.1.0, and GenomicDataCommons v1.15.0. In addition, the tools sourcetracker v2.0.1, the CIBERSORT website (now available as CIBERSORTx[https://cibersortx.stanford.edu/]), FlowJo v10.8.1, and GraphPad Prism 10 were used. Code to regenerate all figures and analyses is available at https://github.com/spakowiczlab/atbac or using the https://zenodo.org/badge/latestdoi/271302970.

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

## Acknowledgements

Funding for this manuscript includes ADA 1-16-1CTS-049 to W.A.H. and D.B., R01 HL135622 and R56AI157202 to W.A.H., 1K01AG070310 to D.J.S. We want to thank Kayla Diaz (clinical research manager) for recruiting the patients for this study and the Comprehensive Cancer Center at the Ohio State University for their flow cytometry core and related resources.

## Author contributions

D.S. performed mouse and human flow cytometry experiments and analysis, human gene expression analysis and edited the manuscript; R.H., O.S., and D.J.S. processed and analyzed the data and edited the manuscript; A.M.B. collected and performed the human data and analyses; L.A., Y.L., and C.W. processed and analyzed the data; A.J. collected some of the human data and edited the manuscript; V.P.W. performed the mouse experiments; J.L. participated in human adipose tissue fractionation, adipocyte gene expression, and edited the manuscript; A.J.S. performed some of the flow analyses and edited the manuscript; D.B. performed statistical analyses and edited the manuscript; W.L. and L.S.S. provided valuable guidance; N.W. generated the microbiome data; B.N., S.B., S.N., D.R. and K.A.P. provided surgical human adipose tissue biopsies for the study; P.N. edited the manuscript; D.W., S.M., and P.R. performed some of the neutrophil flow data; M.P. performed ampliseq analyses; D.J.S. directed and supervised the microbiome and bioinformatic aspects of the project; W.A.H. supervised the project and experiments and oversaw data analyses and manuscript preparation.

## Competing interests

The authors declare no competing interests.
