## [Peer Review File · Nature Communications]

Obesity-associated microbiomes instigate visceral adipose tissue inflammation by recruitment of distinct neutrophilsREVIEWER COMMENTS

Reviewer #1 (Remarks to the Author):

The manuscript submitted by Blaszcak et al evaluated the role of neutrophils in adipose tissue (AT) inflammation and obesity. The authors use a set of interesting mouse experiments to show the impact of a high fat diet on intestinal permeability and neutrophil recruitment. Further, the authors also tried to verify their findings in humans by using RNAseq analyses and deconvolution of publicly available datasets. Therefore, the manuscript actually consists of two parts, an experimental mouse part and a more descriptive human part. Although both parts are interesting, many conclusions from the mouse part could not be reproduced in humans. Although the data are interesting, some conclusions need better controlled experiments and proper discussion as listed in detail below.

Major points:

1. Since the experimental setup of mouse studies only includes a three day (!) HFD feeding period, it is inaccurate to use this as a model for obesity (otherwise almost all peoples in the US and Europe would fulfill criteria for “obesity” after Christmas holidays). It would be more accurate to talk about acute effects of HFD feeding, which maybe also explains differences to human data (see below).
2. In other obesity mouse models, there is also an acute neutrophil recruitment after onset of the HFD feeding period (PMID: 18503031, 22863787, 28202548). However, later in time, lymphocytes and macrophage recruitment occurs as a result of neutrophil-driven inflammation. Therefore, the verification of a later AT inflammation in HFD-treated mice with obese microbiota would be necessary to confirm the pathophysiological mechanism in instigating AT inflammation.
3. The microbiome transplantation experiments were only performed using one lean (BMI 21) and one obese fecal sample (BMI 31). If this is correct, the data need to be reproduced by more fecal samples to verify general validness.
4. In human data sets, differences between lean and obese VAT neutrophils are marginal as stated by the authors. Therefore, the authors pooled data from lean and obese humans for further analyses, whereas in mice, the impact of human microbiome was essential for

neutrophil recruitment and activation. Therefore, the impact of obesity and intestinal barrier dysfunction for human neutrophils is doubtful. In line, commercially available zonulin ELISAs are well-known to be unspecific and are not useful to assess intestinal permeability, which needs to be considered and discussed (PMID: 31563879; 33037053).

5. Next, the specific VAT-signature of neutrophils assessed in VAT-neutrophils from lean and obese AT, fails to be VAT specific, since many other tissues (liver, muscle, breast, stomach, kidney, adrenals, esophagus, thyroid, uterus.....), which are not blood or blood-filled (such as the spleen) show a similar predominance of “VAT-neutrophils” over blood derived neutrophils. Therefore, this signature more likely reflects a tissue phenotype of neutrophils rather than a VAT-specific signature (maybe this could be tested in the mouse model with labeled neutrophils before and after recruitment to different tissues). In summary, in the perspective of this reviewer, there is no obesity- or adipose-context for a unique neutrophil activation within the human data.

6. In line 201, the authors state that “increase in genes related to bactericidal activity and LPS-induced genes indicate that these neutrophils had come into contact with bacteria”. However, there are no significant differences in regard to gene expression for bactericidal activity in the related Fig. 4B. In fact, there are not even clear trends.

Minor points:

1. Have the authors any evidence for an influence of antibiotics on intestinal permeability as describes earlier (PMID: 26691591; 31211803; 33841420).
2. What are the dominating taxa in the saline control group, which should reflect endogenous microbiota after antibiotic washout?
3. In the abstract in line 73, please omit “while on a HFD”.
4. In fig. 2C/D labelling of the x-axis is unreadable.
5. Please add a scale to the PC plots in Fig. 4A to visualize the distribution of data.
6. The correlations from Fig. 4C are unclear to this reviewer. Please explain again.
7. Please provide the gating strategy and flow plots for visualization of the respective immune populations.

Reviewer #2 (Remarks to the Author):

The article entitled “High-fat diet and obese gut microbiome instigate visceral adipose tissue inflammation by recruitment of a distinct type of neutrophil” is addressing the important role of VAT neutrophils in obesity, suggesting a link between gut microbiota and neutrophil recruitment and VAT inflammation. In human samples, the authors observe a higher proportion of neutrophils in VAT of obese individuals and a more pro-inflammatory profile, as previously described. They also observed an association between the proportion of neutrophils in VAT and inflammatory/obesity markers, and a specific perivascular localization (Fig. 1). The authors next studied the adipose tissue microbiota of some lean and obese individuals and observed differences between obese and lean VAT (family Streptococcaceae and a genus in the family Ruminococcaceae enriched in obese VAT versus the genus Marvinobryantia and the order Bacilliales in lean AT) (Fig. 2). Using a model of gut microbiota transfer in mice by gavage, the authors next studied the recruitment of neutrophils only in animals receiving HFD and gut microbiota from obese individuals and showed a double role of the regimen and the gut microbiota (Fig 3). They next perform transcriptomic analyses to characterize the VAT neutrophils, that appear to differ from blood neutrophils and showed similarity with endotoxin activated neutrophils (Fig 4). Lastly, they use transcriptome dataset to identify a specific VAT neutrophil signature and evaluate their presence in various tissues and showed a similar profile notably in tumor tissues, suggesting a role for VAT neutrophils in obesity-related tumors (Fig 5).

Although the work is of interest, I believe that some aspects are limiting its potential and some conclusions are not fully supported by the data. As a whole, the data presented are mostly observational. My main criticisms are the following

1- for the human studies, the proportion of male and female individuals are clearly different and sex/gender may modulate neutrophils biology (Gupta, PNAS, 2020 for example). Is the difference maintained when comparing only male or only female?

2- How AT was dissociated? It is not mentioned in the article, but the difference between blood and AT neutrophils may also differ due to the differences in treatment. It is an important point as the VAT neutrophils signature may mostly reflect technical treatment impacting the neutrophil subsets. In that respect, finding similarity with tumor infiltrating neutrophils may support this notion of a tissue-specific signature, due to tissue-specific

treatment to isolate immune cells. Due to this aspect, I was more cautious when reading the Fig4 and Fig5 data. Could it be a technical artefact?

3- For the experimental setting using the gut microbiota transfer by gavage (Fig. 3), the rationale for this setting is not clear to me. Why only a three days HFD versus standard chow regimen? (Is it the expected timing for neutrophil recruitment? but then are the authors expecting a transient increase in neutrophils or a consistent increase). This experiment is not fully conclusive in its setting as it does not provide clear result on the contribution of the diet regimen versus gut microbiota on neutrophils recruitment. Does the absence of bacteria grafting in SC animals reflect a too short time delay? Could it be a difference in kinetics? Is there any difference in VAT composition and/or inflammatory profile between HFD and standard chow at day 3? Does this neutrophil recruitment persist? Also the number of samples in this experiment is limited, which may also explain the limitation in interpretation.

4- The link between VAT neutrophils (whose specific signature may be discussed) with tumor neutrophils is observational and is not directly proven.

5- Numbers of samples are not always clearly indicated in the manuscript.

In more details, my comments are the following

Figure 1

- The difference in neutrophil percentage is observed on two groups on lean and obese individuals, with different sex-ratio. Could the authors confirm that the higher proportion of female in the obese group is not impacting the results?
- How AT was dissociated? How adipocytes and stromal vascular fraction were collected?
- Why analyzing the inflammatory profile on "Visceral adipocytes"? It sounds appropriate for adiponectin and leptin that are mostly produced by adipocytes, but why using such strategy for the other cytokines, mostly produced by immune cells? The analysis of the SVF would be more appropriate for this (as performed in Fig. 3).
- How many samples were studied for the PCR part and the microscopy part?

Figure 2

- How many samples were tested? (is it 10 obese and 7 lean individuals, as suggested by the graphs in C and D ? If so, how the individuals were selected (VAT neutrophil count? IMC?)
- What is the quantity of bacterial load found in obese and lean AT? Is there any difference?

(And what happen in terms of diversity?)

- Is there any association between VAT neutrophil count and AT microbiota composition?
- How is performed the Bayesian estimation of community source?

Figure 3

- How many animals were tested? How many feces were tested? How the selection was made?
- Is it statistically relevant?
- I'm not sure to read properly the figure 3B: where are the standard chow conditions for the saline and lean contexts? Is it below detection limit? (and is the detection limit different for each tissue (brain/liver/VAT)?
- Is there any change in VAT composition and/or inflammation profile in the saline group, that may help to identify the change associated with the HFD versus SC regimen?
- Could the authors discuss these differences in microbiota composition compared to other AT microbiota studies ?

Figure 4

- The analysis is performed on 9 obese and 6 lean individuals. Are they equivalent in terms of sex ratio, age, and for the obese, what is the IMC?
- Did the authors check the profile of neutrophils after the isolation of the SVF from VAT, and the sort? Could the authors estimate the level of activation induced by the isolation /sort?
- Are the differences shown in Fig4.B significant or not?

Figure 5

- Why pooling the data from lean and obese individuals? It may create a bias for the subsequent analysis
- No comparison with bone marrow neutrophils available?
- the authors showed an association between "VAT" neutrophils in tumor and survival based on RNASeq data from colon adenocarcinoma. However, it relies on the existence of a specific VAT neutrophils signature, that is not fully supported by the data, as previously discussed. However, the link between neutrophils profile and cancer survival is of obvious

interest, as already documented (Hedrick, Nat Rev I 2022 for review, for example). This part is interesting but not fully investigated, which minor the potential of the article.

Reviewer #3 (Remarks to the Author):

This is a fascinating and well designed study. The combination of human characterization and prospective testing in the mouse model is compelling.

The authors evaluated a sizable human cohort subsetted based on leanness and obesity. Upon examination, the authors found increased neutrophilia in visceral adipose tissue of obese individuals relative to lean individuals. Hypothesizing that a combination of the gastrointestinal microbial community and an individual's diet may orchestrate recruitment of neutrophils into VAT, the authors assembled a cohort of mice that had previously been ablated of microorganisms and gavaged them with fecal matter collected from either a lean individual or obese individual (one limitation here is that only one donor for leanness and one for obesity was used; all downstream comparisons are more reflective of differences between two individuals than the are of "leanness" and "obesity"). The gavaged mice were fed either normal cow or a high-fat diet.

The authors found increased bacterial diversity in visceral adipose tissue of all mice fed a high-fat diet, regardless of whether or not they were gavaged with fecal matter or saline, whereas mice fed normal chow did not show this increased diversity. Interestingly, both increased bacterial and neutrophil infiltration was observed only in mice fed a high-fat diet and gavaged with fecal matter from an obese individual.

Minor technical issues aside, these results alone are fascinating and open up avenues for additional inquiry.

Consider the following...

- Switch to qPCR using universal 16S primers to accurately obtain absolute abundance of the microorganisms in the examined compartments
- Contemporaneously sampled potential source locations (gut, lung, etc.) would have been

helpful in ascertaining the origin of the VAT microbes.

- Figure 1A and C: show individual replicates instead of columns

- Figure 1I: reduce size of red arrows; they are huge

- Figure 2C and D: check axis labels

- add to your discussion any steps that you took to identify and remove the confounding effects of reagent contaminants from your analysis.

REVIEWER COMMENTS

Reviewer #1 (Remarks to the Author):

The manuscript submitted by Blaszczyk et al. evaluated the role of neutrophils in adipose tissue (AT) inflammation and obesity. The authors use a set of interesting mouse experiments to show the impact of a high-fat diet on intestinal permeability and neutrophil recruitment. Further, the authors also tried to verify their findings in humans by using RNAseq analyses and deconvolution of publicly available datasets. Therefore, the manuscript actually consists of two parts: an experimental mouse part and a more descriptive human part. Although both parts are interesting, many conclusions from the mouse part could not be reproduced in humans. Although the data are interesting, some conclusions need better controlled experiments and proper discussion, as listed in detail below.

Major points:

R1.1: Since the experimental setup of mouse studies only includes a three-day (!) HFD feeding period, it is inaccurate to use this as a model for obesity (otherwise, almost all peoples in the US and Europe would fulfill criteria for “obesity” after Christmas holidays). It would be more accurate to talk about acute effects of HFD feeding, which maybe also explains differences to human data (see below).

Response: *We agree that the mouse studies are not a model of obesity. They are a model to demonstrate requirements for neutrophils to collect in VAT, namely HFD in the recipient mice, which causes increased epithelial transport and gavage with obese, but not lean, human microbiomes.*

R1.2. In other obesity mouse models, there is also an acute neutrophil recruitment after onset of the HFD feeding period (PMID: 18503031, 22863787, 28202548). However, later in time, lymphocytes and macrophage recruitment occurs as a result of neutrophil-driven inflammation. Therefore, the verification of a later AT inflammation in HFD-treated mice with obese microbiota would be necessary to confirm the pathophysiological mechanism in instigating AT inflammation.

Response: *Thank you for the suggestion. In the studies cited, mice were maintained on prolonged HFD, which could have elicited changes in the other immune cell types independent of neutrophils, or the study was not continued beyond 3 days. We placed mice on normal chow diet after the fecal gavage and 5 days of HFD to differentiate effects of HFD vs. neutrophils. Although the neutrophil peak subsided during the normal chow period, a rise in VAT Th1 cells and a decrease in Tregs occurred at 10 days after gavage. No macrophage changes were seen during this time.*

R1.3: The microbiome transplantation experiments were only performed using one lean (BMI 21) and one obese fecal sample (BMI 31). If this is correct, the data need to be reproduced by more fecal samples to verify general validity.

Response: *We repeated the experiment with stool samples from two additional lean and four obese individuals, so we have a total of 2 lean and 3 obese female donors and 1 lean and 1 obese male donors. Overall, the samples behaved similar to the first obese and lean females. The samples from obese donors increased AT neutrophils, while samples from lean did not (Figure 3, Supplementary Figures 8 and 9).*

R1.4: In human data sets, differences between lean and obese VAT neutrophils are marginal as stated by the

authors. Therefore, the authors pooled data from lean and obese humans for further analyses, whereas in mice, the impact of human microbiome was essential for neutrophil recruitment and activation. Therefore, the impact of obesity and intestinal barrier dysfunction for human neutrophils is doubtful. In line, commercially available zonulin ELISAs are well-known to be unspecific and are not useful for assessing intestinal permeability, which needs to be considered and discussed (PMID: 31563879; 33037053).

Response: *We appreciate the ability to clarify. Lean human VAT neutrophil expression is very similar to obese VAT neutrophil expression but the NUMBER of VAT neutrophils is greatly increased in obese which is a major finding in our study. We have added additional text to clarify this point. Although we report little transcriptomic differences between VAT neutrophils from lean and obese individuals, there were major differences between VAT and blood neutrophils, with VAT neutrophils being more inflamed than blood. We then used mice to demonstrate that the cause of the increased VAT neutrophils was both obese fecal microbiome (no increase in neutrophils with lean fecal microbiome) and HFD (no increase in VAT bacteria or neutrophils with chow diet). Several previous studies have shown that HFD induces a “leaky gut.” However, to pursue this issue further, we administered HFD to 11 healthy lean individuals and observed increased serum zonulin after two weeks of HFD (these data are now added to Figure 3).*

Major contributions of our study include 1) the finding of increased VAT neutrophils in human obesity, 2) major transcriptomics differences between VAT and blood neutrophils but not lean and obese neutrophils, 3) two features of obesity, microbiome, and HFD contribute to VAT neutrophilia, as shown by our mouse model, 4) neutrophilia leads to adipose tissue CD4+ T cell changes that are known to occur in obese mice and in obese humans and humans after two weeks HFD (Deng, Nature Comm 2017; Bradley, Nature Comm, 2022)^{1,2}, 5) the finding that neutrophils in other tissues resemble those in VAT and may be sentinel (poised to activation, but not fully activated) to protect against injury, and in fact, predict poor outcomes in colon cancer when their numbers are low. We have added text to clarify these points, which are all novel. We believe the HFD pulse affects gut leakiness, not obesity, per se, and propose that obese individuals chronically eat more HFD, leading to the effects observed in Figure 1.

R1.5: Next, the specific VAT-signature of neutrophils assessed in VAT-neutrophils from lean and obese AT, fails to be VAT specific, since many other tissues (liver, muscle, breast, stomach, kidney, adrenals, esophagus, thyroid, uterus.....), which are not blood or blood-filled (such as the spleen) show a similar predominance of “VAT-neutrophils” over blood derived neutrophils. Therefore, this signature more likely reflects a tissue phenotype of neutrophils rather than a VAT-specific signature (maybe this could be tested in the mouse model with labeled neutrophils before and after recruitment to different tissues). In summary, from the perspective of this reviewer, there is no obesity- or adipose context for a unique neutrophil activation within the human data.

Response: *Yes, we agree. Likely, the **number** of tissue neutrophils and perhaps not their transcriptional difference sets the stage for tissue inflammation. Our mouse model did not see an increase in lung, liver, or brain microbes or neutrophils, suggesting VAT may be more susceptible to microbe uptake than other tissues. Our data are consistent with a previous report suggesting uptake and efflux of tissue neutrophils in normal mice³. Although defining the neutrophil presence in other tissues is outside the scope of our study, in Figure 6G, the VAT-like signature occurred only in a subset of neutrophils in lung cancer, suggesting the presence of VAT-like neutrophils is not a general tissue phenomenon.*

R1.6: In line 201, the authors state that “increase in genes related to bactericidal activity and LPS-induced genes indicate that these neutrophils had come into contact with bacteria.” However, there are no significant differences in regard to gene expression for bactericidal activity in the related Fig. 4B. In fact, there are not even clear trends.

Response: *We have clarified the text and figure to emphasize that **all** the genes shown in Figure 5B (previous figure 4B) significantly differ between VAT and PB. In addition, we adjusted the figure to re-scale the y-axis with each panel so the significant differences between the groups are easier to see.*

Minor points:

R1.7: Have the authors any evidence for an influence of antibiotics on intestinal permeability as describes earlier (PMID: 26691591; 31211803; 33841420).

Response: *In the experiment presented in Figure 3, all mice received antibiotics and antifungals. However, the chow-fed mice did not show bacterial translocation into VAT. We interpret this result to suggest that the HFD is the primary driver of intestinal permeability, rather than the antibiotics or antifungals.*

R1.8: What are the dominating taxa in the saline control group, which should reflect endogenous microbiota after antibiotic washout?

Response: *We have added this information to the result section of the manuscript. In addition, we stratified the median counts by whether the mice received a HFD or normal chow, to better illustrate the difference between these experimental groups. While the source of the microbes is not clear – as it is not the gavage – HFD strongly affects the amount of microbes in VAT. The dominant microbes in VAT following the saline gavage were *Ralstonia pickettii*, *Delftia acidovorans*, *Rhodococcus erythropolis qingshengii* sp5959.*

R1.9: In the abstract in line 73, please omit “while on a HFD”.

Response: *Omitted, thank you.*

R1.10: In fig. 2C/D labeling of the x-axis is unreadable.

Response: *We enlarged the x-axis label.*

R1.11: Please add a scale to the PC plots in Fig. 4A to visualize the distribution of data.

Response: *We have added the % variance captured by each principle component and included marginal distributions for each axis (now figure 5A.).*

R1.12: The correlations from Fig. 4C are unclear to this reviewer. Please explain again.

Response: *We have added text to clarify the correlations and several other aspects of this figure (now Figure 5), which we agree is not a standard visualization. However, we felt this was the best way to succinctly capture the overlap in the differentially expressed genes between the different publicly available neutrophil datasets. We chose an “Upset”-style plot (https://en.wikipedia.org/wiki/UpSet_Plot) to show Venn-diagram-type overlaps with many groups. The correlation coloring is a custom addition to this plot to indicate the expression similarity (Pearson correlation) between the shared genes. In this case, the DEGs for each of the six datasets are compared; the bars at the top of the plot indicate the number in each group, where the groups are defined according to the dots in the center of the panel. For example, the leftmost, tallest bar is the “VAT alone” category (single dot, not connected to other dots), with > 1000 significant DEGs. Its category circle is colored purple, indicating a correlation of 1 because the single category is compared against itself. In this version of the plot, we drew a red box around the bar and group indicator circles for the “VAT-Sepsis” category to draw the readers’ attention to the height of this bar (indicating more shared DEGs than any other pair of neutrophil groups), and the strongest correlation between the expression levels of the VAT and Sepsis groups (dark green color, correlation ~0.4), further indicating that these groups are the most similar.*

R1.13: Please provide the gating strategy and flow plots to visualize the respective immune populations.

Response: *We added text to the Methods and included Supplementary Figures 1,6 and 7.*

For Human: Neutrophils were defined as Lin- (CD3, CD14, CD19, CD20, CD56), CD16+, CD11b+, and CD15+. They were also further characterized by markers of activation and migration: CD66b and CD62L, respectively. All antibodies were purchased from Biolegend unless otherwise specified. Fixable viability dye was used to exclude dead cells from analysis.

For mice: for flow cytometric analysis of mouse VAT and splenic neutrophils- (Lin- CD3, CD19, CD14), CD45+, CD11b+ Ly6G+, CD64-, CD11c-. Macrophages- (Lin- CD3, CD19, NK1.1), CD45+, CD11b+, CD64+, F4/80+. Regulatory T cells (Tregs)- (Lin- CD19, Gr-1, NK1.1), CD45+, CD3+ CD4+ CD25+, Foxp3+. T helper 1 (Th1) cells- (Lin- CD19, Gr-1, NK1.1), CD45+, CD3+ CD4+ CD25-, Tbet+ CXCR3+. All antibodies from Biolegend, CA, USA were used. Representative gating strategies demonstrated in Supplementary Figures 6 and 7.

Reviewer #2 (Remarks to the Author):

The article entitled “High-fat diet and obese gut microbiome instigate visceral adipose tissue inflammation by recruitment of a distinct type of neutrophil” is addressing the important role of VAT neutrophils in obesity, suggesting a link between gut microbiota and neutrophil recruitment and VAT inflammation. In human samples, the authors observe a higher proportion of neutrophils in VAT of obese individuals and a more pro-inflammatory profile, as previously described. They also observed an association between the proportion of neutrophils in VAT and inflammatory/obesity markers, and a specific perivascular localization (Fig. 1). The authors next studied the adipose tissue microbiota of some lean and obese individuals and observed differences between obese and lean VAT (family Streptococcaceae and a genus in the family Ruminococcaceae enriched in obese VAT versus the genus Marvinobryantia and the order Bacilliales in lean AT) (Fig. 2). Using a model of gut microbiota transfer in mice by gavage, the authors next studied the recruitment of neutrophils only in animals receiving HFD and gut microbiota from obese individuals and showed a double role of the regimen and the gut microbiota (Fig 3). They next perform transcriptomic analyses to characterize the VAT neutrophils, that appear to differ from blood neutrophils and showed similarity with endotoxin activated neutrophils (Fig 4). Lastly, they use transcriptome dataset to identify a specific VAT neutrophil signature and evaluate their presence in various tissues and showed a similar profile notably in tumor tissues, suggesting a role for VAT neutrophils in obesity-related tumors (Fig 5).

Although the work is of interest, I believe that some aspects are limiting its potential and some conclusions are not fully supported by the data. As a whole, the data presented are mostly observational. My main criticisms are the following:

R2.1. for the human studies, the proportion of male and female individuals are clearly different and sex/gender may modulate neutrophils biology (Gupta, PNAS, 2020 for example). Is the difference maintained when comparing only male or only female?

Response: We agree that this is an important consideration. We re-analyzed the data in Figure 1 (See Supplementary Figure 3) and show similar results in females alone with a suggestion in males alone (although there were less male samples). We see no significant difference between lean vs obese in the male population, whereas we do see differences in the female population. We note that the heavy skewing toward female, obese participants allows for very little power to detect this difference. By calculating Cohen's d for the female population alone, we estimate 17% power to detect a difference in the male population with this sample size. Therefore, the male population is underpowered, but it is difficult for us to obtain enough males at this time because substantially fewer males undergo bariatric surgery.

R2.2. How AT was dissociated? It is not mentioned in the article, but the difference between blood and AT neutrophils may also differ due to the differences in treatment. It is an important point as the VAT neutrophils signature may mostly reflect technical treatment impacting the neutrophil subsets. In that respect, finding similarity with tumor infiltrating neutrophils may support this notion of a tissue-specific signature, due to tissue-specific treatment to isolate immune cells. Due to this aspect, I was more cautious when reading the Fig4 and Fig5 data. Could it be a technical artefact?

Response: *We agree that neutrophil isolation could impact their phenotype. We have included additional details of the isolation methods and additional experiments to test the effects of the isolation on gene expression. The results are summarized in Supplementary Figure 11 comparing VAT neutrophil expression across three different neutrophil isolation procedures. The figure is shown below:*

Supplementary Figure 11. qRT-PCR validation of VAT neutrophil signature genes compared to peripheral blood (PB) neutrophils (n=5). In order to test the robustness of the VAT neutrophil signature validation, PB neutrophils were isolated by flow sorting, the commonly used dextran method, and subjected to VAT neutrophils isolation conditions (i.e. dissociation buffers, temperatures etc.). All conditions showed similar levels of gene expression. All data represented as mean \pm SD compared by one-way ANOVA followed by post-hoc Tukey's multiple comparisons test. *: p<0.05, **: p<0.01, ***: p<0.001, ****: p<0.0001

PB sort: Flow sorting CD15+ CD11b+ peripheral blood (PB) neutrophils identical to VAT neutrophil isolation method.
PB Dextran: dextran sedimentation is one of the most widely utilized techniques in neutrophil purification from human blood; this method was used for generation of the Ampliseq data.

PB AT protocol: To eliminate the effect of VAT specific reagents on gene expression, we treated PB neutrophils similarly as VAT neutrophils (with dissociation buffers, temperatures, incubations etc.) prior to RNA isolation. We compared the results of three different PB neutrophils isolation methods and found that the Isolation method does not impact the gene expression results in PB neutrophils. (Supplementary Figure 11).

R2.3. For the experimental setting using the gut microbiota transfer by gavage (Fig. 3), the rationale for this setting is not clear to me. Why only a three days HFD versus standard chow regimen? (Is it the expected timing for neutrophil recruitment? but then are the authors expecting a transient increase in neutrophils or a consistent increase). This experiment is not fully conclusive in its setting as it does not provide clear result on the contribution of the diet regimen versus gut microbiota on neutrophils recruitment.

Response: When mice are placed on HFD, neutrophils transiently peak at 3-5 days (see R1.2. references). We compared a standard chow regimen to HFD: only HFD resulted in increased VAT bacteria and neutrophils (Figure 3), indicating that bacterial translocation induced by HFD is necessary. This was a consistent finding using 3 samples of fecal microbiome from lean subjects and 4 from obese subjects. We did not continue HFD because this would allow continuous translocation. Five days were used because this is within the time frame for the neutrophil peak seen with HFD, which we have also noted. We then found that although bacteria increased in VAT in mice ingesting HFD, only the obese samples induced neutrophil accumulation in VAT. Thus, bacteria in obese microbiome initiate VAT inflammation through neutrophils.

R2.4. Does the absence of bacteria grafting in SC animals reflect a too short time delay? Could it be a difference in kinetics?

Response: Our current model and several previous studies indicate that high fat diet leads to gut barrier dysfunction. It's possible that microbes translocate following gavage even in the context of SC diet, but we have no evidence that's the case.

R2.5. Is there any difference in VAT composition and/or inflammatory profile between HFD and standard chow at day 3?

Response: Yes, the immune cell composition changes (see Figures 3C and D), which is verified by PCR in Figure 3E. We have modified the figure and added text to clarify this point.

R2.6. Does this neutrophil recruitment persist? Also the number of samples in this experiment is limited, which may also explain the limitation in interpretation.

Response: We agree that this is a very important consideration and have included an additional experiment to address this concern. With a single fecal gavage, the neutrophil recruitment does not persist beyond week 1.

However, a week later proinflammatory T cell changes occur, which does not happen if we eliminate the neutrophil response with an anti-Ly6G neutrophil depletion antibody (Figure 4). Thus, our mouse model suggests that obese HFD induces bacterial translocation, obese but not lean microbiome induces VAT neutrophil accumulation, and neutrophils induce T cell changes in VAT. We did not see macrophage changes during this time.

R2.7. The link between VAT neutrophils (whose specific signature may be discussed) with tumor neutrophils is observational and is not directly proven.

Response: We agree but felt the results were sufficiently intriguing that the observation should be reported to the community, as suggested by R3 comments. In this version of the manuscript, we explored this relationship further using publicly available scRNAseq data. We found VAT-type expression in a subset of the tumor neutrophils, which we felt strengthened the case that some but not all tumor neutrophils share characteristics of the VAT-type and warrant further investigation.

R2.8. Numbers of samples are not always clearly indicated in the manuscript.

Response: We carefully reviewed the text to ensure the sample size for each analysis is indicated.

In more details, my comments are the following

Figure 1

R2.9. The difference in neutrophil percentage is observed on two groups on lean and obese individuals, with different sex-ratio. Could the authors confirm that the higher proportion of female in the obese group is not impacting the results?

Response: We included an additional Supplementary figure showing results for females separately from males (Supplementary Figure 3). Obese and lean females showed the same difference; data was suggestive in males, but the numbers of males was relatively small. Using the standardized effect size from the female population, this sample size in males has 17% power to detect a difference between lean and obese. As noted in R2.1 above, obtaining male, lean VAT samples is challenging at our institution.

R2.10. How AT was dissociated? How adipocytes and stromal vascular fraction were collected?

Response: Fresh omental VAT (5-20g) is processed within 30 minutes of procurement. 1/2 g is flash-frozen, maintained sterile at -80° C until use for 16s RNA metagenomics. Using a collagenase digestion buffer, minced VAT is digested for 45 min at 37°C in a shaking incubator. Filtration and centrifugation after digestion separates top adipocyte layer and bottom stromal vascular fraction (SVF) pellet. SVF pellet is then subjected to RBC lysis step and prepared for flow staining and analyses. RNA from adipocytes and whole VAT is isolated and stored. We have previous publications for mouse and human VAT collection^{1,2}. Detailed methods have been added to the Methods section as well.

R2.11. Why analyzing the inflammatory profile on “Visceral adipocytes”? It sounds appropriate for adiponectin and leptin that are mostly produced by adipocytes, but why using such strategy for the other cytokines, mostly produced by immune cells? The analysis of the SVF would be more appropriate for this (as performed in Fig. 3).

Response: We have shown that upon HFD, the adipocytes changes from a metabolic cell to an immune cell with proinflammatory functions including antigen presentation^{1,2} (Deng, Nature Comm 2017; Bradley, Nature Comm, 2022). Therefore, analyzing adipocytes provides further insight into VAT inflammation. In addition, we show a 10-fold increase in expression of IL-8, a potent neutrophil chemoattractant.

R2.12. How many samples were studied for the PCR part and the microscopy part?

Response: For Figure 1, inflammatory genes were assessed by qRT-PCR, in obese (n=46) compared to lean (n=10) visceral adipocytes (VAd) in C. For microscopy, 6 obese and 4 lean samples were reviewed in detail, and representative images are shown in I.

Figure 2

R2.13. How many samples were tested? (is it 10 obese and 7 lean individuals, as suggested by the graphs in C and D? If so, how the individuals were selected (VAT neutrophil count? IMC?)

Response: Yes, these numbers are correct. These individuals were selected as the next available to provide a large enough sample for 16s analyses under sterile conditions and SVF flow analyses.

R2.14. What is the quantity of bacterial load found in obese and lean AT? Is there any difference? (And what happens in terms of diversity?)

Response: We attempted to measure the absolute quantity of microbes for these samples but, unfortunately, failed due to technical issues. We see no differences in the VAT microbe quantities in our mouse experiments but appreciate that this model may not recapitulate the human data in this regard. We note no difference in microbe richness or diversity between the obese and lean samples, which were much higher than the negative controls. We included this information as Supplementary Figure 5.

R2.15. Is there any association between VAT neutrophil count and AT microbiota composition?

Response: Unfortunately, we do not have the VAT neutrophil count data from most of the samples with 16S data.

R2.16. How is performed the Bayesian estimation of community source?

Response: The method uses sequenced communities from many reference samples from different sources to develop a Bayesian prior for assigning particular taxa to a given source. The priors for the presence of each taxon are then aggregated to a likelihood for each source. Additional details have been added to the methods to describe this process, including the tool (sourcetracker2) and reference datasets.

Figure 3

R2.17. How many animals were tested? How many feces were tested? How the selection was made?

Response: We included the group sample size in the figure, as opposed to just having the information in the figure legend. There were 5-10 animals/group. Additional details for the criteria used to select stool samples has been included in the Methods. See Supplementary Table 1; briefly, none of the patients had diabetes or other chronic illnesses and were not taking either antihypertensive or anti-cholesterol drugs. Restated here, there are 7 stool samples in this updated manuscript version, 2 females and one male with a BMI of 19-22 and 3 females and 1 male with BMI 30-44. They are generally of similar ages.

Donor ID	Age	Sex	BMI	Lean/Obese	Hypertension/ Dyslipidemia/other	Medications
T6291	37	F	21	L	NO	Birth control
T6824	24	F	31.1	O	NO	NO
T5631	23	M	21.9	L	NO	NO
T6821	26	M	34.96	O	NO	NO
T6557	33	F	18.79	L	NO	Cetirizine, Nexplanon
T6820	26	F	30.45	O	NO	NO
T7154	31	F	43.6	O	NO	NO

R2.18. I'm not sure to read properly the figure 3B: where are the standard chow conditions for the saline and lean contexts? Is it below detection limit? (and is the detection limit different for each tissue (brain/liver/VAT)?

Response: Chow diet for Saline and Lean gavage were not tested in this experiment. We felt that the other controls effectively describe these groups, and therefore felt the use of more animals was not justified. The detection limit is 2 microbe genome copies per ul.

R2.19. Could the authors discuss these differences in microbiota composition compared to other AT microbiota studies?

Response: We added additional discussion to address this point. In particular, we expanded the section on the mouse VAT experiments with identification of pseudomonas, which was previously reported to be found in a number of human tissues⁴.

Figure 4

R2.20. The analysis is performed on 9 obese and 6 lean individuals. Are they equivalent in terms of sex ratio, age, and for the obese, what is the IMC?

Response: In this revised version of the manuscript, we included a table of the demographics, including sex, age, and BMI. See Supplementary Table 5.

R2.21. Did the authors check the profile of neutrophils after the isolation of the SVF from VAT, and the sort? Could the authors estimate the level of activation induced by the isolation /sort?

Response: We have included additional data to test the activity changes associated with technical features of neutrophil isolation (Supplementary figure 11). The expression for a number of genes is extremely similar whether PB neutrophils are isolated by flow sorting, dextran, or sorted after collagenase, etc which is added during VAT isolation. Please see figure above in response to R2.2.

R2.22. Are the differences shown in Fig4.B significant or not?

Response: Now Figure 5. Yes, for all of the genes shown the expression is different between VAT and PB. We adjusted this figure to better show the separation, and included text to emphasize that we're showing only the significantly different genes in the various pathways.

Figure 5

R2.23. Why pooling the data from lean and obese individuals? It may create a bias for the subsequent analysis

Response: We tested the performance of the signature including a separation between lean and obese, but it was not able to correctly identify known VAT and blood neutrophils in the "gold-standard" test experiments (i.e., the results shown in Figure 5C&D for the signature that performed well). The method relies on strongly distinct populations, and our tests indicated that the lean and obese were insufficiently separated.

R2.24. No comparison with bone marrow neutrophils available?

Response: Studies have shown that in obesity, inflammatory mediators like TNF- α , IL-1 β , IL-6, and IL-8 increase bone marrow granulopoiesis^{5,6}, releasing neutrophils from the bone marrow to the peripheral circulation. Moreover, these inflammatory mediators induce de-margination of neutrophils from endothelial walls, resulting in neutrophilia⁷. In our study, We compared VAT neutrophils to peripheral blood neutrophils from the same individuals and found them to have a distinctly different signature (Figure 6). Thus, we think that the peripheral neutrophils are the likely immigrants from the bone marrow that are released in the periphery in an obese state, and they differ from VAT neutrophils.

R2.25. the authors showed an association between "VAT" neutrophils in tumor and survival based on RNASeq data from colon adenocarcinoma. However, it relies on the existence of a specific VAT neutrophils signature, that is not fully supported by the data, as previously discussed. However, the link between neutrophils profile and cancer survival is of obvious interest, as already documented (Hedrick, Nat Rev I 2022 for review, for example). This part is interesting but not fully investigated, which minor the potential of the article.

Response: We are careful to suggest that the VAT-like neutrophil signature we generated may reflect tissue infiltration, as opposed to specifically VAT-tissue infiltration. However, we note that the most common methods of RNAseq deconvolution, which is often used for cancer analyses, rely on the blood-based neutrophil signature, so our findings add value to the field. In addition, we note several genes strongly expressed in our

signature are different from tumor neutrophil datasets. We agree that the cancer relationship is not fully investigated, but feel it is outside the scope of the article to do so. We have included additional details on the known roles of neutrophils in cancer in this version of the discussion. In addition, we performed an additional analysis of publicly available tumor scRNAseq data to test the similarity of the VAT-derived signature to the tumor-based expression.

Reviewer #3 (Remarks to the Author):

This is a fascinating and well-designed study. The combination of human characterization and prospective testing in the mouse model is compelling.

The authors evaluated a sizable human cohort subsetted based on leanness and obesity. Upon examination, the authors found increased neutrophilia in visceral adipose tissue of obese individuals relative to lean individuals. Hypothesizing that a combination of the gastrointestinal microbial community and an individual's diet may orchestrate recruitment of neutrophils into VAT, the authors assembled a cohort of mice that had previously been ablated of microorganisms and gavaged them with fecal matter collected from either a lean individual or obese individual (one limitation here is that only one donor for leanness and one for obesity was used; all downstream comparisons are more reflective of differences between two individuals than the are of "leanness" and "obesity"). The gavaged mice were fed either normal cow or a high-fat diet.

The authors found increased bacterial diversity in visceral adipose tissue of all mice fed a high-fat diet, regardless of whether or not they were gavaged with fecal matter or saline, whereas mice fed normal chow did not show this increased diversity. Interestingly, both increased bacterial and neutrophil infiltration was observed only in mice fed a high-fat diet and gavaged with fecal matter from an obese individual.

Minor technical issues aside, these results alone are fascinating and open up avenues for additional inquiry.

Consider the following...

R3.1. Switch to qPCR using universal 16S primers to accurately obtain absolute abundance of the microorganisms in the examined compartments

Response: *We appreciate this suggestion and agree that this would have been very useful. For the human studies, we tried, but did not get the absolute quantity for technical reasons. We included absolute quantity information whenever possible, including for all of the mouse studies.*

R3.2. Contemporaneously sampled potential source locations (gut, lung, etc.) would have been helpful in ascertaining the origin of the VAT microbes.

Response: *While we weren't able to obtain these data in the human subjects, we attempted this analysis in our mouse studies to answer your question. See Venn diagram relating the VAT microbiome to the gut and lung microbiomes by 16S. The gut contributes to VAT microbes, but the lung likely also contributes. Studies of the lung are warranted but beyond the scope of this investigation.*

R3.3. Figure 1A and C: show individual replicates instead of columns

Response: *Thank you for the suggestion. In this manuscript version, we updated Figure 1 to show all the data points.*

R3.4. Figure 1I: reduce size of red arrows; they are huge

Response: *We decreased the size of the red arrows in Figure 1I.*

R3.5. Figure 2C and D: check axis labels

Response: Thank you for catching this. The axes are now corrected.

R3.6. add to your discussion any steps that you took to identify and remove the confounding effects of reagent contaminants from your analysis.

Response: Thank you for the suggestion. We went to great lengths to prevent and control for contamination and included additional details in the discussion. We employ sterile collection and processing procedures of all tissues, every sample is processed individually, major buffers are not reused—they made fresh for each human sample or mouse experiment, and the flow machine cleaned and calibrated after each use so no contaminating cells.

References

1. Deng T, Liu J, Deng Y, et al. Adipocyte adaptive immunity mediates diet-induced adipose inflammation and insulin resistance by decreasing adipose Treg cells. *Nature communications*. 2017;8(1):15725.
2. Bradley D, Smith AJ, Blaszczak A, et al. Interferon gamma mediates the reduction of adipose tissue regulatory T cells in human obesity. *Nature communications*. 2022;13(1):1-12.
3. Kolaczowska E, Kubes P. Neutrophil recruitment and function in health and inflammation. *Nature reviews Immunology*. Mar 2013;13(3):159-75. doi:10.1038/nri3399
4. Anhe FF, Jensen BAH, Varin TV, et al. Type 2 diabetes influences bacterial tissue compartmentalisation in human obesity. *Nature Metabolism*. 2020;2(3):233-242.
5. Rusten LS, Jacobsen FW, Lesslauer W, Loetscher H, Smeland EB, Jacobsen S. Bifunctional effects of tumor necrosis factor alpha (TNF alpha) on the growth of mature and primitive human hematopoietic progenitor cells: involvement of p55 and p75 TNF receptors. 1994;
6. Sieff CA, Niemeyer CM, Mentzer SJ, Faller DV. Interleukin-1, tumor necrosis factor, and the production of colony-stimulating factors by cultured mesenchymal cells. 1988;
7. Suwa T, Hogg JC, English D, Van Eeden SF. Interleukin-6 induces demargination of intravascular neutrophils and shortens their transit in marrow. *American Journal of Physiology-Heart and Circulatory Physiology*. 2000;279(6):H2954-H2960.

REVIEWERS' COMMENTS

Reviewer #1 (Remarks to the Author):

The manuscript by Blaszczyk et al has gained in clarity and quality due to intensive revision. The reproduction of obese gavage in 3 out of 4 obese samples and the comparison of the VAT signature following different isolation protocols were very important. Further, to clarify that this model only reflects an acute challenge with a HFD which may have (or may not) implications for chronic obesity is important. Therefore, I have no general concerns against publication. However, this reviewer is still not convinced by the VAT-specificity of the neutrophil signature and recommends re-phrasing. Further, although the zonulin ELISA shows expected results in Fig.3I, critical references should be cited and discussed (PMID: 31563879; 33037053).

Reviewer #2 (Remarks to the Author):

I would like to congratulate the authors for the change/addition they provided for this novel version of the manuscript.

- The novel title (even if it's a slight change) appears to me more representative of the work performed. It is also less misleading regarding a potential VAT specific signature, that may mostly refer to a tissue-signature.
- Another concern was the emphasis given to the comparison of the "VAT-like" signature of neutrophils to those observed in tumor samples. I think this aspect is now more balanced. I felt there was too much emphasis on this aspect as the data that were mostly observational.
- The addition of some more samples for the feces transplantation is a clear enhancement.
- The addition of data related to the gender differences in terms of VAT neutrophils proportion is also a great (and intriguing) addition, as well as the control experiments on the impact of tissue dissociation.

As a minor point, I still don't understand why the analysis of the pro-inflammatory profile of VAT is performed solely on the adipocytes fraction. I agree that adipocytes are important producers of pro-inflammatory cytokines, but the SVF is also an important source. The cited papers demonstrate (nicely) that adipocytes per se contribute to AT inflammation, but it

does not exclude an additional (or even synergistic role) with the SVF cells. What is the opinion of the authors on the matter ? Do they believe that measuring the inflammation induced by adipocytes may be a better read out of AT local inflammation ? Anyway, this aspect is more out of curiosity.

Congratulations

REVIEWER COMMENTS

Reviewer #1 (Remarks to the Author):

R1.1 The manuscript by Blaszczak et al has gained in clarity and quality due to intensive revision. The reproduction of obese gavage in 3 out of 4 obese samples and the comparison of the VAT signature following different isolation protocols were very important. Further, to clarify that this model only reflects an acute challenge with a HFD which may have (or has not) implications for chronic obesity is important. Therefore, I have no general concerns against publication. However, this reviewer is still not convinced by the VAT-specificity of the neutrophil signature and recommends re-phrasing.

We appreciate the enthusiasm for the modified manuscript. We maintain that the signature is developed from an isolated VAT- neutrophil, and therefore, we hesitate to use a nomenclature that implies more generality (e.g., tissue-like neutrophil, non-peripheral blood neutrophil, etc.). We note the other reviewer felt the label used in the revised manuscript was an improvement. In this version, wherever applicable to the expression signature, we altered the text to "VAT-isolated neutrophil (VINs) signature" instead of "VAT-type neutrophil signature." We felt like this tweak to mention the source was less presumptive about the state of the neutrophils.

R1.2 Further, although the zonulin ELISA shows expected results in Fig.3I, critical references should be cited and discussed (PMID: 31563879; 33037053).

Thank you for pointing these out. We added these references as part of results and the discussion.

Reviewer #2 (Remarks to the Author):

I would like to congratulate the authors for the change/addition they provided for this novel version of the manuscript.

- The novel title (even if it's a slight change) appears to me more representative of the work performed. It is also less misleading regarding a potential VAT specific signature, that may mostly refer to a tissue-signature.

We appreciate this comment, and attempted to retain the spirit of that change when reducing the number of words to fit the journal's conventions.

- Another concern was the emphasis given to the comparison of the "VAT-like" signature of neutrophils to those observed in tumor samples. I think this aspect is now more balanced. I felt there was too much emphasis on this aspect as the data that were mostly observational.

Thank you. We note that this has been further softened to VAT-isolated neutrophil (VIN) signature.

- The addition of some more samples for the feces transplantation is a clear enhancement.

- The addition of data related to the gender differences in terms of VAT neutrophils proportion is also a great (and intriguing) addition, as well as the control experiments on the impact of tissue dissociation.

We thank the reviewer for their comments on the revised manuscript and appreciate the role they played in improving the manuscript.

As a minor point, I still don't understand why the analysis of the pro-inflammatory profile of VAT is performed solely on the adipocytes fraction. I agree that adipocytes are important producers of pro-inflammatory cytokines, but the SVF is also an important source. The cited papers demonstrate (nicely) that adipocytes per se contribute to AT inflammation, but it does not exclude an additional (or even synergistic) role with the SVF cells. What is the opinion of the authors on the matter? Do they believe that measuring the inflammation induced by adipocytes may be a better read out of AT local inflammation? Anyway, this aspect is more out of curiosity.

We appreciate this comment and agree that our and others' previous data point to a role for both the adipocytes and SVF cells in inflammation. We didn't assess the SVF in the analyses described in Figure 1 because the SVF is a heterogenous mixture of cell types. We would be left without additional information on which cell type was responsible. Instead, we chose to address the role of adipocytes in Figure 1 and the flow-sorted cell fractions in the rest of the manuscript. We have carefully reviewed the text to ensure this logic is clarified. In addition, we added text to the discussion on the relative contributions of adipocytes and SVF.